# Identification of evolutionarily conserved regulators of muscle mitochondrial network organization

Prasanna Katti [1], Peter T. Ajayi [1], Angel Aponte[1], Christopher K. E. Bleck [1] & Brian Glancy [1,2] ✉

Mitochondrial networks provide coordinated energy distribution throughout muscle cells. However, pathways specifying mitochondrial networks are incompletely understood and it is unclear how they might affect contractile fiber-type. Here, we show that natural energetic demands placed on *Drosophila melanogaster* muscles yield native cell-types among which contractile and mitochondrial network-types are regulated differentially. Proteomic analyses of indirect flight, jump, and leg muscles, together with muscles misexpressing known fiber-type specification factor *salm*, identified transcription factors *H15* and *cut* as potential mitochondrial network regulators. We demonstrate *H15* operates downstream of *salm* regulating flight muscle contractile and mitochondrial network-type. Conversely, *H15* regulates mitochondrial network configuration but not contractile type in jump and leg muscles. Further, we find that *cut* regulates *salm* expression in flight muscles and mitochondrial network configuration in leg muscles. These data indicate cell type-specific regulation of muscle mitochondrial network organization through evolutionarily conserved transcription factors *cut*, *salm*, and *H15*.

Proper mitochondrial network formation and maintenance are crucial for cellular energy distribution, cell signaling, and movement of ions, metabolites, mtDNA, and proteins[1–6]. Mitochondrial network structure is highly variable across cell types due to differences in the amount of cellular volume allotted to mitochondria, the size and shape of individual mitochondria, and the configuration or location of mitochondria within the cell, with each of these structural parameters influencing the relative efficiency of interaction and communication among mitochondria and other cellular structures[7–10]. While regulators of cellular mitochondrial volume (e.g., PGC-1, ERR, and PPAR isoforms) and individual mitochondrial size (e.g., Drp1, Mfn1/2, and Fis1) have been widely identified and studied across cell types[11–23], how mitochondrial network configuration is determined as part of the cellular design process is less well understood[24–26].

Mature striated muscles form relatively stable mitochondrial networks[27] comprised of many physically and electrically connected mitochondria[5,28,29], and muscle mitochondrial networks display differences in mitochondrial content, size, and configuration depending on the energetic and contractile force requirements of a given muscle cell type[5,28,30–32]. In mammalian systems, muscle type is commonly classified by both contractile (i.e., fast- or slow-twitch) and metabolic (i.e., glycolytic or oxidative) types[33,34] with contractile type generally defined by myosin isoform composition or myofibrillar ATPase activity[35–37] and metabolic type often inferred based on mitochondrial content or enzyme activity[35,38–40]. Due to the intimate structural and functional relationships between the contractile[41] and mitochondrial networks, it is not surprising that many of the well-known factors involved in muscle fiber type specification can regulate the design of both the contractile and metabolic machineries within the muscle cell[42–45]. Conversely, there are also many examples of alterations in muscle metabolism, mitochondrial content, and mitochondrial size without affecting contractile fiber type[11,15,46–56] demonstrating that

[1]National Heart, Lung, and Blood Institute, National Institutes of Health, Bethesda, MD 20892, USA. [2]National Institute of Arthritis and Musculoskeletal and Skin Diseases, National Institutes of Health, Bethesda, MD 20892, USA. ✉e-mail: brian.glancy@nih.gov

muscle metabolism and mitochondrial structure can be regulated independently of contractile type.

In order to identify regulators of muscle mitochondrial network configuration, particularly with respect to their impact on contractile type, our first aim was to establish a model system allowing for rapid screening of potential regulatory genes or proteins. Unfortunately, commonly used muscle cell culture models (e.g., myoblasts or myotubes) feature underdeveloped contractile networks permitting frequent mitochondrial movement around the cell which makes these systems ineffective for assessment of regulators of the configuration of the relatively static mitochondrial networks observed in adult muscles[27]. As a result, we turned to the genetically tractable fruit fly, *Drosophila melanogaster*, where mitochondrial structure and metabolism have been widely studied in the adult indirect flight muscles[15,46,47,54,55,57–60]. Though contractile characteristics have been assessed across many different adult *Drosophila* muscles[61–64], much less information is available regarding metabolism or mitochondria in adult *Drosophila* muscles beyond the flight muscles[50,65]. Indeed, in contrast to mammalian systems, muscle cell type in *Drosophila* muscles has been largely defined by contractile (i.e., fibrillar or tubular) or electromechanical (i.e., synchronous or asynchronous) properties rather than metabolic characteristics[63,66–69]. Though the indirect flight muscles are the largest and most well-studied muscles in *Drosophila*, the fibrillar (i.e., comprised of individual myofibrils) and asynchronous (i.e., one calcium cycle results in tens of muscle contractions) nature of these muscles are unlike any known mammalian muscle. Conversely, the majority of *Drosophila* muscles are tubular[70] (i.e., form cross-striated myofibrillar networks[32]) and synchronous (i.e., one calcium cycle per contraction) more similar to mammalian skeletal muscles. Despite being classified as the same contractile type, tubular muscles can have different contractile characteristics (e.g., mechanical force/power[61–64] and sarcomere[71] or myofibrillar network[32] structure) consistent with the variable functions of each specific *Drosophila* muscle. We hypothesized that, like mammals, the wide variety of contractile demands faced across *Drosophila* tubular muscles would also necessitate cells of varying oxidative or glycolytic natures, and in turn, muscles with different mitochondrial network configurations. Thus, we aimed to identify *Drosophila* muscles with differing mitochondrial network structures and use this multi-muscle system to screen for genetic regulators of mitochondrial network configuration. Moreover, we sought to determine whether any newly identified genes regulate mitochondrial network configuration in coordination with contractile type (i.e., muscle converts between fibrillar and tubular types) or independently of contractile type (i.e., muscle remains fibrillar or tubular).

Here, we take advantage of the natural energetic and contractile differences[61,63] among muscle types within the genetically tractable fruit fly, *Drosophila melanogaster*, together with the known *Drosophila* muscle type specification factor, *spalt major* (*salm*)[50,67] to identify evolutionarily conserved regulators of muscle mitochondrial network organization. By performing a proteomic screen on muscles with five different combinations of contractile type, mitochondrial network type, and Salm expression level, we identified 142 proteins associated with muscle fiber type. Further, we demonstrate that *H15* is a regulatory transcription factor downstream from *salm* which can independently regulate contractile type or mitochondrial network configuration in a muscle type-specific manner. Moreover, we show that transcription factor *cut* operates upstream of *salm* and can also independently regulate mitochondrial network configuration in a muscle type-specific manner. Finally, we demonstrate that the specification of mitochondrial network configuration in muscles can be regulated separately from cellular mitochondrial volume and individual mitochondrial volume. Our findings suggest that evolutionarily conserved transcription factors including *cut*, *salm*, and *H15* regulate mitochondrial network configuration in muscle cells through a specification process which can operate independently of contractile type, mitochondrial content, and mitochondrial size.

## Results

### Wild-type mitochondrial network and contractile types

To monitor mitochondrial and contractile network morphology in the different muscle types in *Drosophila*, we used the genetically encoded mitochondrial matrix GFP[72] with the UAS-Gal4 system[47] together with F-actin staining (Phalloidin). The indirect flight muscles (dorsal longitudinal muscles (DLMs)) in wild-type adult flies showed the characteristic fibrillar contractile phenotype of many individual myofibrils[67,73] together with many thick, elongated mitochondria interspersed in parallel to the myofibrils (Fig. 1a–d,u) (Supplementary Movie 1). Flight muscles also had relatively low sarcoplasmic reticulum (SR) content as assessed with genetically encoded KDEL-RFP (Fig. 1e, v and Fig. S1a–c). On the other hand, the jump (tergal depressor of the trochanter (TDT)) muscles exhibited closely aligned myofibrils representative of their tubular contractile phenotype combined with fewer thin, elongated mitochondria also arranged in parallel to the axis of muscle contraction (Fig. 1f–i, u) (Supplementary Movie 2) and high SR content (Fig. 1j, v and Fig. S1d–f). In the leg muscles (coxa of the forelegs), we consistently observed different phenotypes in three spatially distinct regions we termed Fibers I, II, and III (Fig. 1k–q). All three leg regions displayed tubular myofibrils (Fig. 1m, o, q, and Fig. S2). However, Fiber I had primarily parallel mitochondrial networks (Fig. 1l,u) (Supplementary Movie 3) and lower mitochondrial content (Fig. 1t) compared to the more grid-like networks in Fibers II and III (Fig. 1m, n, p, u) (Supplementary Movies 4 and 5). All three leg regions had higher SR content than the flight muscles (Fig. 1v and Fig. S1g–r). The grid-like nature of the mitochondrial networks in the tubular leg muscles (Fiber II and III, Fig. 1r), the parallel mitochondrial networks in the fibrillar flight (Fig. 1d) and tubular jump muscles (Fig. 1i), and the relative differences in SR content were confirmed by focused ion beam scanning electron microscopy (FIB-SEM, Fig. 1e, j, s). These data from wild-type fly muscles show that parallel mitochondrial networks can occur together with either fibrillar (flight) or tubular (jump) contractile types and that tubular muscles can have either parallel (jump/leg Fiber I) or grid-like mitochondrial networks (leg Fiber II/III) (Fig. S3), demonstrating that contractile type and metabolic phenotypes can be regulated independently in *Drosophila* muscles.

### Regulators of mitochondrial networks in fly muscles

Mitochondrial dynamics proteins have been implicated in mitochondrial network formation across many cell types[20,22,74–77]. Thus, we initially attempted to alter mitochondrial network configuration by knocking down mitochondrial dynamics proteins which promote mitochondrial fusion (Marf, mfn1/2 ortholog), mitochondrial fission (Drp1 and Fis1), and mitochondrial motility (Miro) in a muscle-specific manner (using *Mef2-Gal4*). Loss of *Marf* is sufficient to induce smaller, more circular mitochondria (Fig. S4c, m, n), lower mitochondrial volume (Fig. S4o), and the complete loss of flight and climbing ability (Fig. S5a, b). However, *Marf* KD did not change the orientation of mitochondrial networks in either the flight or Fiber II leg muscles (Fig. S5d, k, l). Reductions in fission proteins Drp1 and Fis1 to levels sufficient to cause larger, more elongated mitochondria (Fig. S4e, g, m, n) affected neither mitochondrial network configuration (Fig. S4f, h, o) nor flight or climbing ability (Fig. S5a, b). Finally, loss of Miro also resulted in smaller, more circular mitochondria (Fig. S4i, m, n) and a complete lack of flight and climbing ability (Fig. S5a, b), but did not alter mitochondrial network configuration (Fig. S4j, k, l). These results indicate that individual mitochondrial size can be up or down-regulated without altering mitochondrial network configuration in muscle cells.

Next, we examined the role of zinc finger transcription factor *salm* in the regulation of muscle-specific mitochondrial network organization, by performing *salm* misexpression in *Drosophila*

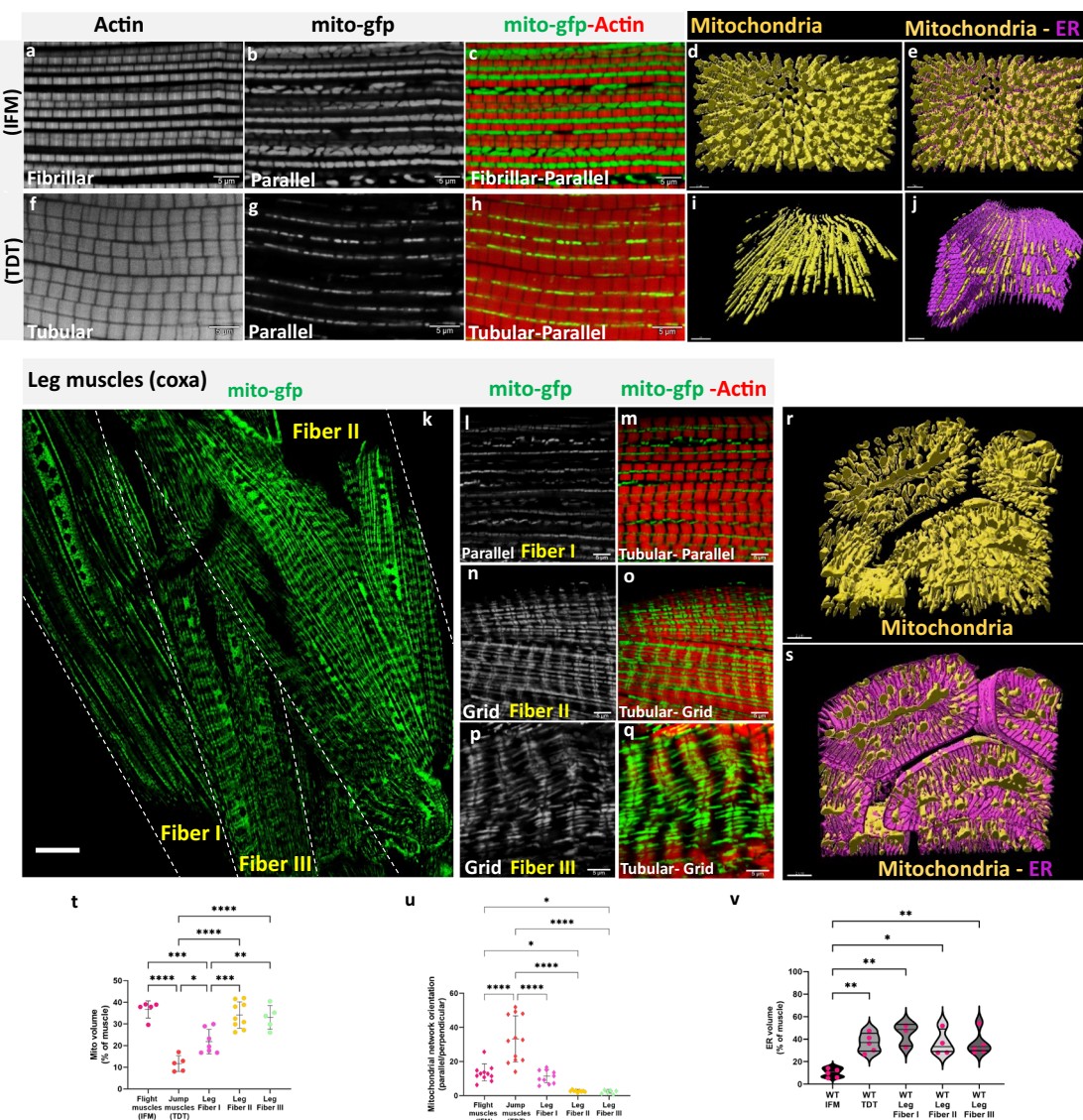

**Fig. 1 | Mitochondrial network organization in adult Drosophila muscles.**
**a**, **b**, **c** Fibrillar flight muscles (IFMs) stained for F - actin and mitochondria expressing mito-gfp driven by *DMef2-Gal4* showing parallel aligned mitochondria that are large, tube-like, and packed between myofibrils (Scale Bars: 5 μm).
**d**, **e** Representative 3D rendering of electron microscopic images of mitochondrial arrangement (yellow) and ER (magenta) in flight muscles. **f**, **g**, **h** Tubular jump muscles show mitochondria that are thin and elongated arranged in parallel mitochondrial networks (Scale Bars: 5 μm). **i**, **j** Representative 3D rendering of mitochondrial networks (yellow) and ER (magenta) in jump muscles. **k** In walking (leg) muscles, *DMef2-Gal4* driven mito-gfp shows both parallel and grid-like mitochondrial networks (marked with dashed line, scale bar: 20 μm). **l**, **m** Tubular leg muscle Fiber I showing primarily parallel mitochondrial networks. **n**, **o** Tubular leg muscle Fiber II showing a grid-like mitochondrial network. **p**, **q** Tubular leg muscle Fiber III showing grid-like mitochondria. (Scale Bars: 5 μm for all).

**r**, **s** Representative 3D rendering of mitochondrial network organization (yellow) and ER (magenta) in leg muscles. **t** Mitochondrial volume as a percent of total muscle volume (Flight muscles (IFM) (*UAS-mito-gfp;Dmef2-Gal4*), $n = 5$ animals; Jump muscles (TDT) (*UAS-mito-gfp;Dmef2-Gal4*), $n = 5$ animals; Leg Fiber I, $n = 7$ animals; Leg Fiber II, $n = 9$ animals; Leg Fiber III, $n = 5$ animals). **u** Quantification of mitochondrial network orientation. Dotted line represents parallel equal to perpendicular (IFM, $n = 11$ animals; Jump muscles (TDT), $n = 12$ animals; Leg Fiber I, $n = 10$ animals; Leg Fiber II, $n = 9$ animals; Leg Fiber III, $n = 6$ animals). **v** Endoplasmic reticulum (ER) volume as a percent of total muscle volume (IFM, $n = 5$ animals; Jump muscles (TDT), $n = 5$ animals; Leg Fiber I, $n = 3$ animals; Leg Fiber II, $n = 4$ animals; Leg Fiber III, $n = 5$ animals). Each point represents value for each animal dataset. Bars represent mean ± SD. Significance determined as $p < 0.05$ from one way ANOVA with Tukey's (*, $p \leq 0.05$; **, $p \leq 0.01$; ***, $p \leq 0.001$; ****, $p \leq 0.0001$; ns, non-significant).

muscles. Salm is a known regulator of fibrillar muscle fate in *Drosophila*[67], and recently was reported to be involved in specifying mitochondrial location in flight muscles[50]. As shown previously[78], *salm* was highly expressed in the wild-type flight muscles (Fig. S6a–d), while lower levels of *salm* expression were observed in wild type jump muscles (Fig. S6e–h) and expression was undetectable in wild type leg muscles (Fig. S6i–l). Muscle-specific RNAi-mediated *salm* knockdown using *Mef2-Gal4* (Fig. S6m–p) resulted in flightless flies with a reduced jumping ability (Fig. S5a, c) and conversion of the flight muscle contractile apparatus to tubular muscle and the mitochondrial networks

to grid-like (Fig. 2a–h,o). It is important to note that *salm* KD mediated conversion of muscle type occurs whether one (*UAS-salm RNAi*, Figs. S6, 7) or two (*UAS-salm RNAi::UAS-mito-GFP*, Fig. 2f–h) UAS are present. Additionally, the fiber transformation in *salm* KD flight muscles was accompanied by an increase in ER content to the level of the wild type tubular muscles (Fig. S7j–l). On the other hand, *salm* KD had no effect on the contractile type or mitochondrial morphology in the jump or leg muscles (Fig. S7d, e, h, and Fig. S8a–d). These results are consistent with *salm* as a critical specification factor between flight and leg muscle phenotypes.

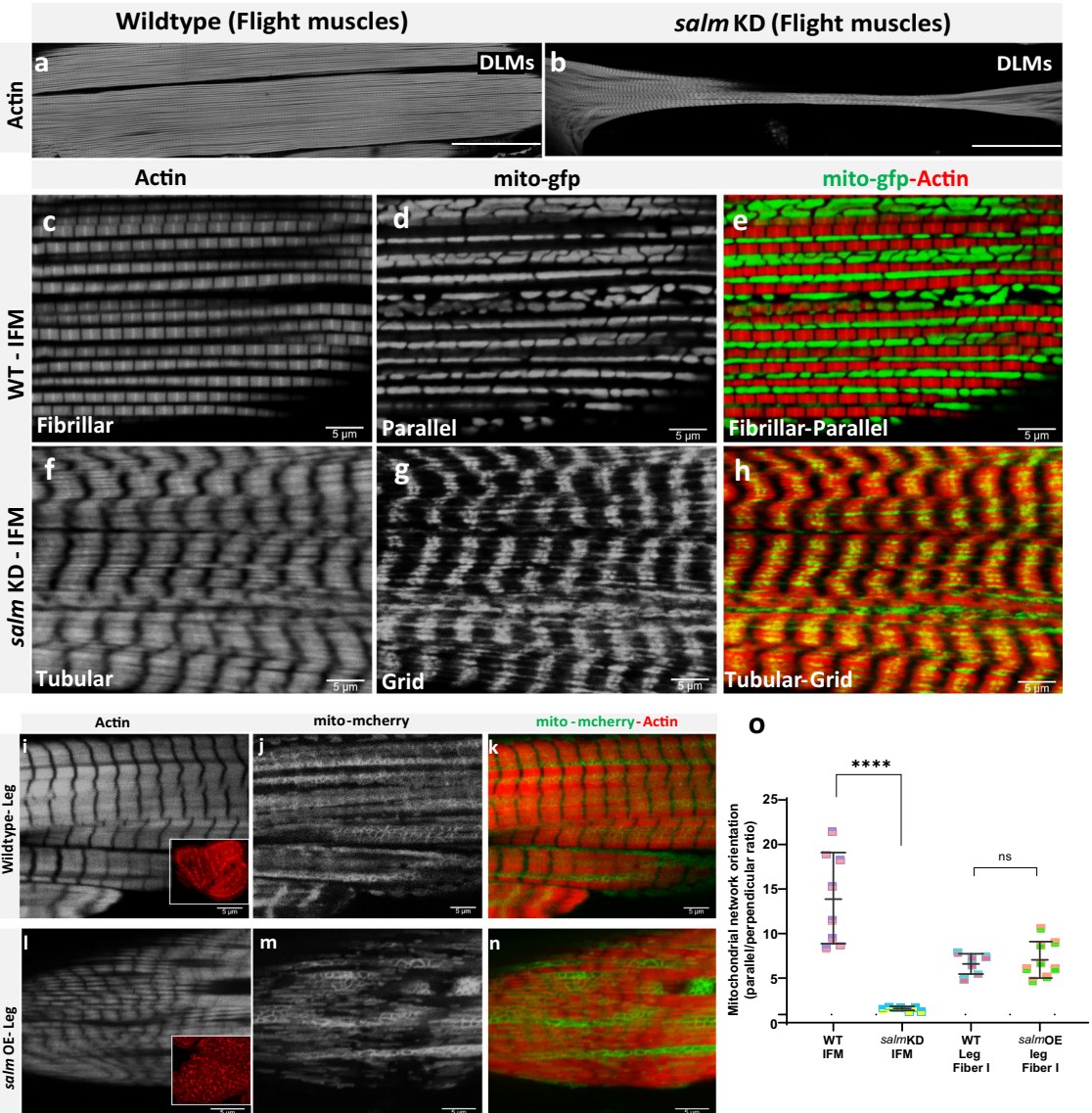

**Fig. 2 | *salm* regulates conversion of muscle fiber contractile type and mito-chondrial network orientation. a** Adult wild-type flight muscles (fibrillar) stained for F-actin (phTRITC). **b** *salm* KD muscle fiber stained for muscles showing tubular muscle type (Scale Bars: 100 μm). **c, d, e** Wildtype flight muscles show elongated, parallel mitochondria (mito-gfp) between myofibers (phTRITC). **f, g, h** The knockdown of *salm* (*UAS-salm RNAi;UAS-mito-gfp;mef2*) in flight muscles results in fiber conversion to tubular muscle type and mitochondria to a grid-like network. (i) Wild type leg muscles show tubular muscle type in the coxa. Inset shows a cross-section of leg muscles displaying well-aligned fibers. **j, k** Wildtype leg muscles show parallel mitochondria (mito-mcherry) aligned next to myofibrils (phTRITC). **l** *salm* OE converts muscle fibers to fibrillar in nature. Inset, well-defined fibrillar fibers in cross-section of leg muscles. **m, n** *salm* OE leg muscles have parallel mitochondrial networks (mito-mcherry) along the myofibrils (phTRITC) (Scale Bars: 5 μm for all). **o** Quantification of mitochondrial network orientation. Dotted line represents parallel equal to perpendicular. *mito-gfp;mito-mcherry;mef2-Gal4* used as Wildtype, WT. (WT-IFM, *n* = 9 animals; *salm*-KD IFM, *n* = 7 animals; WT-Leg Fiber I, *n* = 6 animals; *salm*-OE Leg Fiber I, *n* = 8 animals). Each point represents value for each animal dataset. Bars represent mean ± SD. Significance determined as *p* < 0.05 from one way ANOVA with Tukey's (*, *p* ≤ 0.05; **, *p* ≤ 0.01; ***, *p* ≤ 0.001; ****, *p* ≤ 0.0001; ns, non-significant).

To investigate the impact of increased salm expression on mitochondrial network organization, we overexpressed (OE) *salm* in *Drosophila* muscles (Fig. S6q–t). *salm* OE in muscles resulted in pupal lethality and escapers showed dysfunctional walking behavior (100%, Fig. S5b). Importantly, *salm* OE transformed the tubular leg myofibrils to more fibrillar-like myofibrils similar to those in flight muscles (Fig. 2i, l). However, following *salm* OE, the mito-chondrial networks in the transformed leg muscles remained similar to those in wild type tubular walking muscle fibers as there was no significant difference in the ratio of parallel to perpendicular mitochondria compared to wild type leg muscles (Fig. 2i–o). In the *salm* OE studies, we used Tom20-mcherry instead of mito-GFP as the genetically encoded mitochondrial reporter because *UAS-mito-*

*GFP*, as well as Tub-Gal80ts and *UAS-salm* used for salm OE, are located on the same chromosome. However, no differences in the mitochondrial network configuration were observed across muscle types between mito-GFP and Tom20-mcherry reporters (Fig. S9), suggesting that neither the choice of mitochondrial marker nor the use of a second UAS influenced our network orientation results. Additionally, *salm* OE had no apparent effect on the parallel mito-chondrial networks in the jump muscle or flight muscle (Fig. S8e, g). Overall, these results demonstrate that *salm* is a critical factor in determining mitochondrial network organization in flight muscles and in determining contractile type in both flight and leg muscles, but *salm* does not appear to play a key role in jump muscle con-tractile or metabolic specification.

## Proteomic screen for factors associated with fiber type

To identify additional potential regulators of muscle fiber type specification, we performed a high-throughput, TMT-based[79] proteomic screen on the three wild-type muscles (flight, jump, leg) (Fig. 1) together with the two muscles which underwent fiber type conversion with *salm* misexpression (*salm* KD flight, *salm* OE leg) (Fig. 2). A total of 3869 proteins were quantified for each sample (Supplemental Data 1). While the overall protein abundance profile of each sample was similar (Fig. S10a), principal component and heat map clustering analyses indicated strong reproducibility and clear distinctions among the five muscle types assessed (Fig. S10b, c). To characterize the differences among muscle types, differentially expressed proteins (Fig. S11) were run through a gene enrichment analysis using g:Profiler[80] and primarily identified myofibrillar and mitochondrial processes, consistent with our phenotypic image analyses above (Fig. S12).

To identify regulators specific for each muscle mitochondrial network configuration and/or contractile type, we rationalized that by assessing protein expression in muscles each with varying combinations of mitochondrial network configuration (parallel or grid-like), contractile type (fibrillar or tubular), and *salm* expression (Supplementary Table 1), we could make multiple comparisons of differentially expressed proteins associated with a given phenotype and identify proteins which were consistently associated across all comparisons. For example, we hypothesized that proteins positively associated with fibrillar muscle fate (Fibrillar+) would be higher in wild type flight muscles compared with wild type leg (558 proteins), wild type jump (1326 proteins), or *salm* KD flight muscles (597 proteins) and in *salm* OE leg muscles compared to wild type leg muscles (1387 proteins) (Fig. 3a). Of all the proteins identified by each individual comparison, 25 proteins were consistently found across all four Fibrillar+ comparisons (Fig. 3a). To determine proteins negatively associated with fibrillar muscle fate (Fibrillar-), the inverse analysis was performed by identifying proteins that were lower in the wild type flight muscles compared to wild type leg (2450 proteins), wild type jump (771 proteins), or *salm* KD flight muscles (1674 proteins) and in *salm* OE leg muscles compared to wild type leg muscles (590 proteins) yielding 2 proteins that were consistent across all Fibrillar- comparisons (Fig. 3b). Similar analyses identified 6 Tubular+ proteins (Fig. 3c), 7 Tubular- proteins (Figs. 3d), 3 Parallel+ proteins (Fig. 3e), 23 Parallel- proteins (Fig. 3f), 565 Grid+ proteins (Fig. 3g), 9 Grid- proteins (Figs. 3h), 1 salm+ protein (Fig. 3i), and 50 salm- proteins (Fig. 3j). The relatively large number of Grid+ proteins is due in part to having only three rather than four or five comparisons groups like for the other phenotypes (Fig. 3). As a result, we further filtered the Grid+ proteins down to 16 using the microarray data from Schonbauer et al[67]., which reported transcripts with higher expression levels in two muscles shown here to have grid-like mitochondrial networks, the wild type leg and *salm* KD flight muscles, relative to wild type flight muscles. Thus, in total, we identified 142 proteins whose expression levels consistently correlated with at least one fiber type specification phenotype (Supplemental Data 2).

## *H15* regulates muscle fiber type downstream of *salm*

Of the 51 *salm*-associated proteins, there was only one Salm+ protein, H15, which met the thresholds of our screen. *H15* is a T-box transcription factor, orthologous to the vertebrate proteins, *Tbx20* and *Tbx15*. Importantly, *Tbx15* has been suggested to regulate glycolytic fiber-type specification in a whole-body knockout model in the mouse[81], though how *Tbx15* interacts with other transcription factors to specify muscle cell fate remains unclear. Based on the positive association with *salm* expression, we hypothesized that *H15* would regulate fiber type specification downstream of *salm* in *Drosophila* muscles and provide further support for the evolutionarily conserved nature of the *salm*-associated regulation of mitochondrial network configuration identified in *Drosophila* muscle. To assess the impact of

*H15* on mitochondrial network organization and muscle contractile specification in muscles, we first performed muscle-specific *H15* KD using *Mef2-Gal4* during *Drosophila* muscle development. We initially tested two different *H15* KD lines (V28415, V106875) and found that the V28415 *H15* KD line led to viable flightless adults (Fig. S5a) whereas the V106875 *H15* KD line resulted in weak flight. Thus, we chose the stronger phenotype (V28415) for all further *H15* KD analyses. *H15* KD was confirmed by immunofluorescence (Fig. S18a–h, z) and resulted in contractile type switching of the flight muscle fibers from fibrillar to tubular (Fig. 4a, d) (Fig. S13e–h). Further quantitative qPCR analysis showed significant decrease in fibrillar sarcomeric transcript expression, *Flightin* (fln), *troponin* C (Tpnc4), and *Actin88F* (Fig. S13i). Moreover, mitochondrial content was significantly reduced in the flight muscles following *H15* KD (15.59 ± 1.13% of muscle volume) when compared to the wild-type flight muscles (38.88 ± 0.9%, Fig. S14a), and the mitochondrial networks remained organized parallel to the axis of contraction similar to wild type flight and jump muscles (Fig. 4a–f). Indeed, there was no significant difference in the ratio of parallel to perpendicular mitochondria between *H15* KD and wild-type flight muscles (Fig. 4s). Thus, in contrast to *salm* KD, *H15* KD induced a jump muscle phenotype (tubular/parallel) rather than a leg muscle phenotype (tubular/grid) in the flight muscles indicating that mitochondrial network configuration can be regulated independently from contractile type in the flight muscle.

In the jump muscles, *H15* KD did not appear to affect the tubular nature of the myofibrils (Fig. 4g, j). However, *H15* KD converted the parallel mitochondrial networks to a more grid-like arrangement (Fig. 4g–l) (Supplementary Movie 6) as evident in the decrease in the ratio of parallel to perpendicular mitochondria in *H15* KD jump muscles compared to wild-type jump muscles (Fig. 4s) (Supplementary Movie 2). Thus, loss of *H15* in the jump muscles resulted in a phenotype similar to the wild-type leg muscles (Fiber II, III tubular/grid) and was accompanied by a reduction in jumping ability (Fig. S5c). These results suggest that *H15* regulates mitochondrial network configuration but not contractile type in the jump muscles. However, it should be noted that while *H15* KD does not alter the tubular contractile type of the jump muscles, muscle-specific loss of *H15* has recently been shown to increase sarcomere branching frequency[32] and myosin filament curvature[71] above the level of the wild-type jump muscles and closer to the levels of the wild type leg muscles. Those results are consistent with the interpretation here that *H15* KD in the jump muscle results in a leg muscle-like phenotype based on the tubular contractile type and grid-like mitochondrial networks.

In the leg muscles following *H15* KD, the contractile structure was clearly tubular in all three fiber regions (Fig. S15l, o, r). However, knockdown of *H15* transformed the parallel mitochondrial networks of Fiber I in the leg to a grid-like organization (Fig. 4m, p) (Supplementary Movie 7), as confirmed by the decrease in the ratio of parallel to perpendicular mitochondria compared to the wild type leg muscle Fiber I (Fig. 4t). Conversely, *H15* KD did not affect the grid-like mitochondrial networks of Fiber II and Fiber III (Fig. 4q, r, t), which remained similar to those in the wild type leg muscles (Fig. 4n, o, t). These data further suggest that *H15* regulation of mitochondrial network organization occurs in a cell-type-specific manner and can operate independently of contractile type specification.

To test whether the flight muscle contractile phenotype was specifically due to *H15* down-regulation rather than off-target effects, individual *Mef2*-Gal4 driven knockdowns of 16 genes that are off-targets of the *H15* RNAi line (VDRC 28415) were evaluated based on a previous flight muscle genetic screen[82] and our own imaging of eight different knockdowns (Supplemental Table 2). Downregulation of these off-targets did not cause a tubular conversion of the flight muscles (Fig. S16), though loss of CG31374 (*sals*) led to partial lethality with eclosing flies unable to fly likely related to the apparent shortening and thinning of the sarcomeres as reported previously[83].

## Proteins associated with contractile type

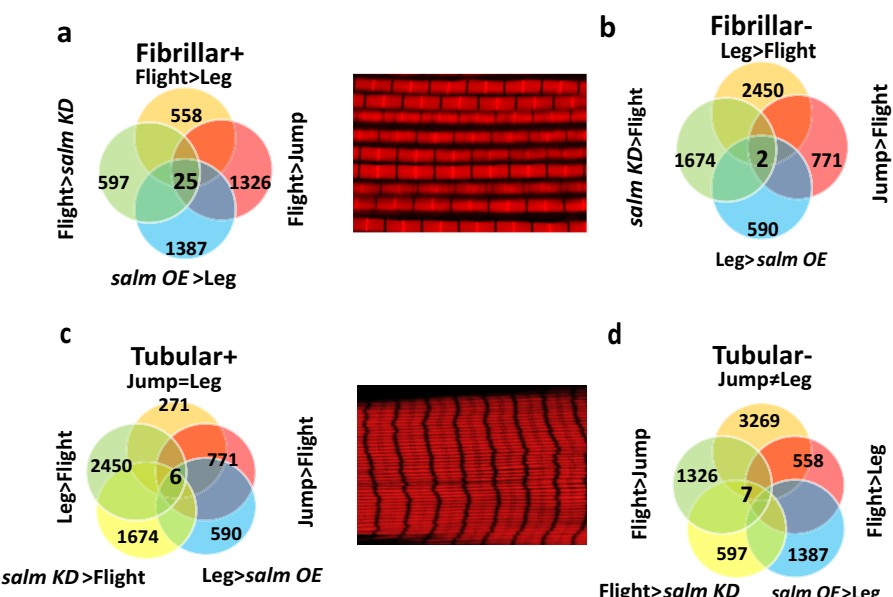

## Proteins associated with mitochondrial network shape

## Salm associated proteins

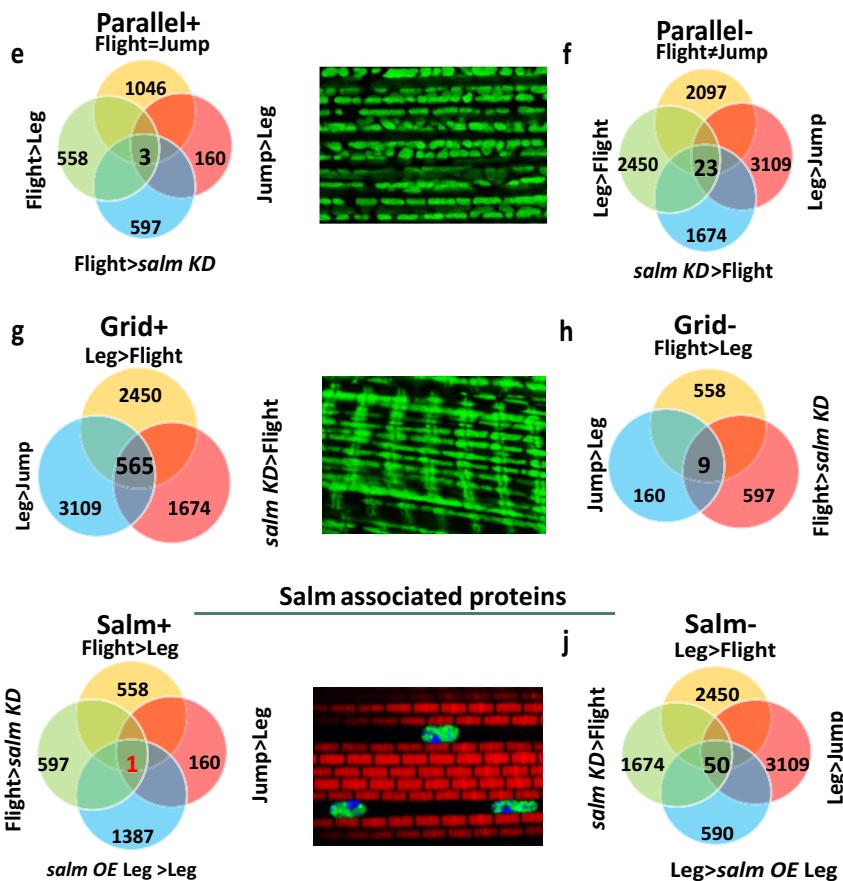

**Fig. 3 | Identification of potential regulatory factors for contractile and mitochondrial network type in Drosophila muscle. a** Venn diagram displaying the individual muscle group comparisons and their overlap for proteins positively associated with fibrillar muscles. **b** Venn diagram displaying the individual muscle group comparisons and their overlap for proteins negatively associated with fibrillar muscles. **c, d** Venn diagram for tubular positively and negatively associated proteins, respectively. **e, f** Venn diagram for positively and negatively associated proteins, respectively, with parallel mitochondrial networks. **g, h** Venn diagram for positively and negatively associated proteins, respectively, with grid-like mitochondrial networks. **i, j** Venn diagram for *salm* positively and negatively associated proteins, respectively.

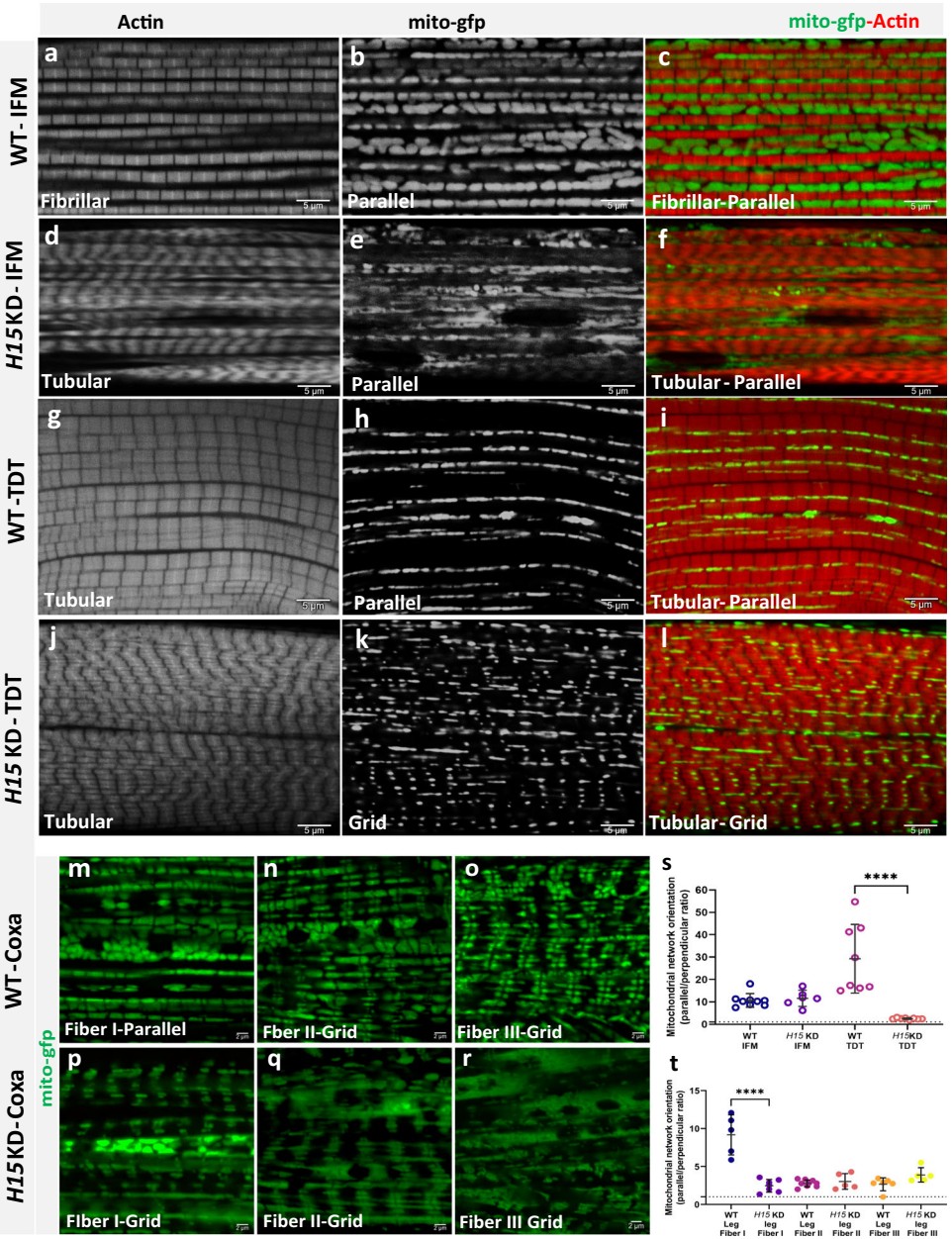

**Fig. 4 | *H15* independently regulates the conversion of muscle contractile type and mitochondrial network configuration in Drosophila muscles.**
**a**, **b**, **c** Fibrillar flight muscles (IFMs) stained for F-actin (phTRITC) and mitochondria (mito-GFP) showing parallel aligned mitochondria between myofibrils. **d** *H15* KD shows fibrillar muscles switched to tubular muscle type. **e**, **f** *H15* KD muscles show parallel mitochondria along muscle fibers. **g**, **h**, **i** Tubular jump muscles show parallel mitochondria (thin and elongated). **j** *H15* KD in jump muscles shows tubular fibers (phTRITC). **k**, **l** Upon *H15* KD, tubular jump muscles show a change in mitochondrial networks to more grid-like (Scale Bars: 5 µm for all). **m** Wildtype coxa leg muscle Fiber I showing parallel mitochondrial networks. **n** Wildtype leg muscle Fiber II and **o** Fiber III showing grid-like mitochondrial networks. **p** *H15* KD leg muscle Fiber I showing conversion to a grid-like mitochondrial network. **q**, **r** *H15* KD

leg muscle fibers II and III show grid-like structures similar to their wild type counterparts (Scale Bars: 5 µm for all). **s** Quantification of mitochondrial network orientation. Dotted line represents parallel equal to perpendicular. *mito-gfp;mito-mcherry;mef2-Gal4* used as wildtype, WT; indirect flight muscles, IFM; Jump muscles, TDT. (WT-IFM, $n = 9$ animals; *H15* KD-IFM, $n = 6$ animals; WT-TDT, $n = 8$ animals; *H15* KD-TDT, $n = 8$ animals). **t** Quantification of mitochondrial network orientation in leg (walking) Fibers. (WT-leg Fibers I, $n = 5$ animals; *H15* KD-IFM, $n = 7$ animals; WT-leg Fiber II, $n = 9$ animals; *H15* KD-Leg Fiber II, $n = 5$ animals; WT-Leg Fiber III, $n = 6$ animals, *H15* KD-Leg Fiber III, $n = 5$ animals). Each point represents value for each dataset. Bars represent mean ± SD. Significance determined as $p < 0.05$ from one way ANOVA with Tukey's (*, $p \le 0.05$; **, $p \le 0.01$; ***, $p \le 0.001$; ****, $p \le 0.0001$; ns, non-significant).

In addition, we also assessed the role of *mid*, a paralog of *H15*, in mitochondrial network organization and muscle contractile specification in muscles and found that *mid* KD did not cause conversion of myofibrillar or mitochondrial networks in flight, jump and leg fiber I muscles (Fig. S17). These results suggest that the muscle type conversions observed in the *H15* KD flies are mediated specifically by *H15* rather than off-target or paralogous genes.

To test our hypothesis that *H15* operates downstream of *salm* in *Drosophila* muscle, we first assessed Salm and H15 expression in wild type, *salm* KD, and *H15* KD flight muscles by immunofluorescence (Fig. S18). As mentioned above, *salm* KD resulted in a loss of Salm expression (Fig. S7, Fig. S20y), and *H15* KD led to a reduction in H15 expression (Fig. S20f, z) in flight muscles as expected. While *salm* KD in flight muscle led to a decrease in H15 protein and mRNA expression

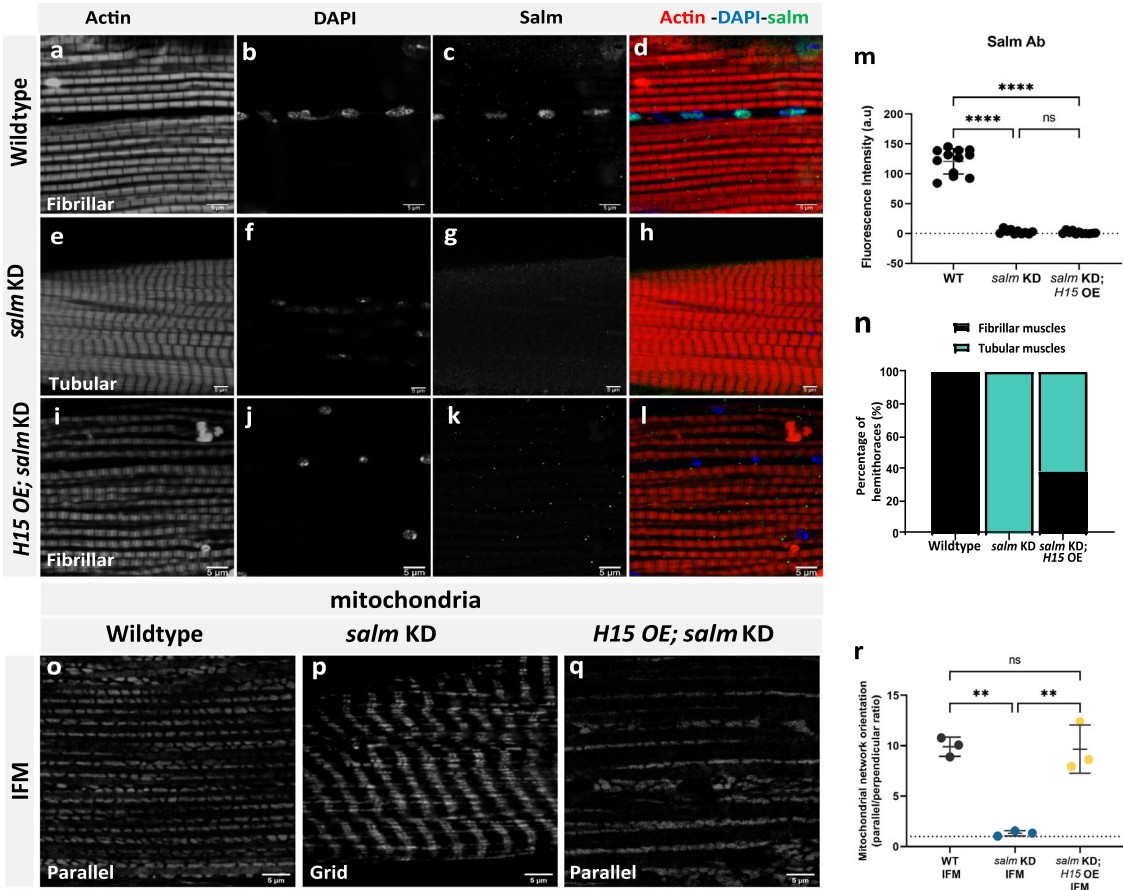

**Fig. 5 | H15 is downstream of salm and H15 OE rescues fiber-type switching in salm KD fibers. a–d** Wildtype fibrillar IFM stained for F-actin (phTRITC), nuclei (DAPI), and Salm antibody showing Salm expression in the nuclei. **e–h** salm KD IFM showing decreased expression of salm and fiber type switch in IFM from fibrillar to tubular. **i–l** H15 OE; salm KD shows fibrillar fiber type in IFMs and decreased Salm antibody immunofluorescence (Scale Bars: 5 µm for all). **m** Quantification of fluorescence intensity of Salm antibody staining (Wildtype, *n* = 12, 3 animals; salm KD, *n* = 11, 3 animals; H15 OE; salm KD, *n* = 11, 4 animals). Each point represents the value for each dataset (Wildtype, *n* = 16; salm KD (*UAS-salm RNAi;UAS-mito-gfp;mef2*) (, *n* = 20 animals; H15 OE; salm KD, *n* = 26 animals). **n** Quantification of fiber type in IFMs (Wildtype, *n* = 16; salm KD (*UAS-salm RNAi;UAS-mito-gfp;mef2*)

(*n* = 20; H15 OE; salm KD, *n* = 26). **o** Wildtype IFM showing a parallel mitochondrial network (MitoTracker). **p** Knockdown of salm (*UAS-salm RNAi;UAS-mito-gfp;mef2*) in flight muscles coverts parallel mitochondrial arrangement to a grid-like network. **q** H15 OE; salm KD shows parallel mitochondrial networks in IFMs similar to wildtype (Scale Bars: 5 µm for all). **r** Quantification of mitochondrial network orientation. Dotted line represents parallel equal to perpendicular (WT IFM, *n* = 3 animals; salm KD IFM, *n* = 3 animals; H15 KD; salm OE IFM, *n* = 3 animals). Each point represents value for each animal dataset. Bars represent mean ± SD. Significance determined as *p* < 0.05 from one way ANOVA with Tukey's (*, *p* ≤ 0.05; **, *p* ≤ 0.01; ***, *p* ≤ 0.001; ****, *p* ≤ 0.0001; ns, non-significant).

(Fig. S18v, z, z'), thereby providing validation to our proteomic findings; *H15* KD had no effect on Salm expression in the flight muscles (Fig. S18n, y) consistent with *H15* regulation of muscle fiber type specification occurring downstream of *salm*.

To provide confirmation of *H15* operating downstream of *salm* in *Drosophila* muscle, we performed rescue experiments where we overexpressed *H15* in the *salm* KD background (Fig. 5 and Fig. S19). Overexpression of *H15* together with *salm* KD (*H15 OE; salm KD*) did not rescue the flightless phenotype caused by the loss of *salm* (Fig. S5a). However, *H15 OE; salm KD* did successfully rescue the fibrillar contractile type of the flight muscles from the tubular contractile type observed in *salm* KD flight muscles (Fig. 5a, e, i) in 38% of the muscle fibers observed (*n* = 26) (Fig. 5m). Salm expression remained undetectable by immunofluorescence in the fibrillar *H15 OE; salm KD* flight muscles (Fig. 5a–l, n) indicating that the contractile type rescue was indeed mediated by *H15*. Moreover, mitochondrial networks (assessed with MitoTracker Red) in the rescued *H15 OE; salm KD* flight muscles also returned to their wild-type, parallel phenotype (Fig. 5o–r), indicating that *H15* can regulate mitochondrial network configuration in flight muscles in the absence of *salm*. This *H15* OE rescue of the *salm* KD phenotype is not due to GAL4 dilution as the *salm* KD phenotype is

retained in the presence of two UAS (*UAS-salm RNAi::UAS-mito-GFP*, Fig. 2f–h). Overall, these data demonstrate that *H15* regulates muscle fiber type specification downstream of *salm*.

## cut regulates mitochondrial network organization

To identify additional regulators of mitochondrial network organization and/or contractile type in *Drosophila* muscle, we focused on the transcription factors in addition to H15, which were associated with at least one muscle specification phenotype in our proteomic screen. Of the 142 candidate proteins, six additional transcription factors (Prospero, Limpet, Reversed polarity, cut, CG17822, and CG12605) were identified using the *Drosophila* Transcription Factor Database[84] including three salm- proteins (Prospero, Reversed polarity, and Limpet) and one each for Fibrillar + (CG12605), Grid + (Limpet), Grid-(CG17822), and Parallel- (cut). We generated flies with muscle-specific (*Mef2-Gal4*) KD or OE for three of these proteins (Prospero, Limpet, and cut) to determine their potential role in muscle fiber type specification.

Limpet is a LIM domain protein that expresses different isoforms in fibrillar and tubular muscles with its splicing pattern regulated by *salm*[85]. Our proteomic analysis detected four different isoforms of

Limpet (A, J, K, and N) (Supplemental Data 1) with isoforms A and P both being identified as Grid+ and Salm- proteins (Supplemental Data 2) consistent with expression of these isoforms being highest in the leg (Supplemental Data 1). Thus, we hypothesized that *Limpet* KD would alter the mitochondrial network and/or contractile type in the legs whereas *Limpet* OE would have the greatest effect on the flight muscles. However, *Limpet* KD flies appeared to fly normally and showed no changes in mitochondrial network configuration or contractile type in the flight, jump, or leg muscles (Fig. S20). Additionally, *Limpet* OE flies also showed the wild-type mitochondrial network and contractile phenotypes across each muscle group (Fig. S21), though these flies displayed weak flight behavior. These data suggest that while Limpet expression is associated with muscle type, *Limpet* itself does not regulate the configuration of contractile or mitochondrial networks in *Drosophila* muscle.

Prospero is a homeobox transcription factor critical for neurogenesis and motor neuron innervation in *Drosophila* muscle[86,87]. Additionally, loss of the mammalian ortholog of Prospero, Prox1, in striated muscle leads to an upregulation of fast-twitch contractile proteins in both the heart and skeletal muscle of mice[88,89], and Prox1 directly interacts with ERRα and PGC-1α to negatively modulate their activity in mouse liver cells[90]. Thus, we hypothesized that *Prospero* would regulate both contractile and mitochondrial network type in *Drosophila* muscles. Knockdown of *Prospero* throughout muscle development (*Mef2-Gal4*) was lethal consistent with the known essential role of its ortholog, Prox1, in myoblast differentiation[89]. *Prospero* KD from the third instar larval stage (*Tub-Gal80[ts] Mef2-Gal4*) allowed flies to reach adulthood though they remained flightless. Despite the loss of overall muscle function with *Prospero* KD, the mitochondrial and contractile networks in the flight, jump, and leg muscles retained their wild-type phenotypes, respectively (Fig. S22). Due to the identification of Prospero as a Salm- protein in our proteomic screen, we hypothesized that *Prospero* OE would convert the contractile and/or mitochondrial networks in the flight muscles. However, flies with muscle-specific overexpression of *Prospero* maintained parallel mitochondrial networks and fibrillar myofibrils similar to wild-type flight muscles (Fig. S23) albeit with weak flight ability. Thus, despite its indispensable role for normal muscle development and function, these results indicate *Prospero* does not regulate contractile or mitochondrial network configuration in *Drosophila* muscles.

*cut* encodes a homeobox transcription factor involved in cell-type specification and patterning across *Drosophila* organ systems[91–94]. In muscle, cut expression level is known to differentiate between which third instar wing disc myoblasts will eventually form the fibrillar indirect flight muscles (low cut) or the tubular direct flight muscles (DFM, high cut)[95,96]. However, the role of *cut* in the development of contractile networks in muscles derived from outside the third instar proximal wing imaginal disc (e.g., jump or leg muscles) or in the specification of mitochondrial network type remains unclear. Due to the known role of *cut* in DFM specification, we first assessed mitochondria in the wild-type DFMs by FIB-SEM which revealed mitochondrial networks arranged in sheets parallel to the tubular contractile networks (Fig. S24) suggesting that *cut* regulates contractile type, but not the orientation of mitochondrial networks in DFMs. Since our proteomic analyses identified cut as a Parallel-protein (Supplemental Data 2), we hypothesized that *cut* KD would alter the grid-like mitochondrial networks in the leg muscles and that *cut* OE would affect the parallel mitochondrial networks in the flight and jump muscles. Muscle-specific (Mef2-Gal4) *cut* KD permitted development to adulthood, but with a complete loss of flight activity and weak climbing ability compared to wild-type flies (Fig. S5a, b). We were unable to detect cut by immunofluorescence in wild-type adult muscles, so muscle-specific *cut* KD was confirmed by the loss of cut immunofluorescence in Mef2-positive third instar proximal wing imaginal disc cells (Fig. S26). Flight and jump muscle contractile and mitochondrial networks were unaffected by *cut* KD (Fig. S27), as were the contractile networks in the leg muscles (Fig. S28a, d, g, j). However, while the mitochondrial networks in Fibers I and III of the *cut* KD legs retained the wild-type parallel and grid-like configurations, respectively (Fig. 6a, b and Fig. S28a–l), the mitochondrial networks in Fiber II of the *cut* KD leg were converted from grid-like to parallel (Fig. 6d, g, i) (Supplementary Movie 8) in 78.6% of the muscles assessed ($n = 14$) (Fig. S28m) without a change in mitochondrial content (Fig. 6j). These results indicate that *cut* regulates mitochondrial network configuration, independent of contractile fiber-type, in a regionally specific manner within the leg muscles.

To identify whether *cut* operates in the same mitochondrial network configuration specification pathway as *salm* and *H15*, we further investigated the relationship between *cut* and *salm*. Expression of cut protein was increased above wild type levels in both the *salm* KD flight muscles and the *salm* OE leg muscles (Supplemental Data 1) suggesting that *salm* does not regulate the expression of *cut*. Indeed, *cut* and transcription factor *vestigial* are known to act in a mutually repressive fashion[95], and *salm* expression in flight muscles has been shown to require upstream expression of *vestigial*[67], suggesting that *cut* may act as an upstream repressor of *salm*. To test this hypothesis, we investigated whether combining *cut* KD and *salm* KD (*cut* KD; *salm* KD) would rescue the contractile and mitochondrial network phenotypes observed in *salm* KD flight muscles. While *cut* KD; *salm* KD flies were unable to fly (Fig. S5a), the flight muscle contractile networks were fibrillar and the mitochondrial networks were parallel similar to the wild type flight muscles (Fig. 7d–f). Further, significant Salm immunofluorescence was detected in the *cut* KD; *salm* KD flight muscles (Fig. 7q) indicating that Salm may no longer be knocked down sufficiently to induce a fiber type transformation. To investigate this possibility, we performed qPCR analysis of *salm* transcript levels in the flight muscles of wild type, *salm* KD, *cut* KD, and *cut* KD; *salm* KD flies. Though *salm* KD resulted in an ~60% decrease in *salm* transcript expression relative to wild-type flies, there was no difference in *salm* expression between wild type and *cut* KD; *salm* KD flight muscles (Fig. 7s) indicating that *salm* was no longer knocked down upon the addition of *cut* KD. The rescue of *salm* expression in the *cut* KD; *salm* KD flight muscles was not due to GAL4 dilution in the presence of a second UAS (*UAS-salm RNAi::UAS-cut RNAi*) since *salm* KD mediated flight muscle conversion still occurs with two UAS (*UAS-salm RNAi::UAS-mito-GFP*, Fig. 2f–h). Further, *salm* transcript levels were increased ~60% above wild type levels in the *cut* KD flight muscles (Fig. 7s) indicating that the increase in *salm* expression in the *cut* KD; *salm* KD flight muscles relative to the *salm* KD flight muscles was indeed mediated by *cut*.

To further test the hypothesis that *cut* operates upstream of *salm*, we assessed the impact of *cut* OE on contractile and mitochondrial network configuration and *salm* expression. Muscle-specific (Mef2-Gal4 driven) overexpression of *cut* resulted in 100% pupal lethality and precluded phenotypic analyses of adult flight and jump muscles. However, flight muscle-specific (Act88f-Gal4 and 1151-Gal4 driven) overexpression of *cut*, confirmed by qPCR (Fig. 7t), allowed adult flies to eclose and resulted in tubular flight muscles with centralized nuclei and grid-like mitochondrial networks similar to those observed in *salm* KD flight muscles (Fig. 7j–l,m, Fig. S25). Moreover, both Act88f-Gal4 and 1151-Gal4 driven *cut* OE resulted in a loss of Salm protein and transcript expression compared to wild-type flight muscles (Fig. 7n–r, t, Fig. S25) similar to the reductions observed in *salm* KD flight muscles (Fig. S7o, Fig. 7s). Additionally, Act88f-Gal4 driven *cut* OE led to a reduction in *H15* transcript expression whereas the 1151-Gal4 driven loss of H15 expression due to *cut* OE did not reach significance (Fig. 7t). Overall, these data demonstrate that *cut* operates as an upstream repressor of *salm* capable of regulating contractile and mitochondrial network configuration in a muscle-specific manner.

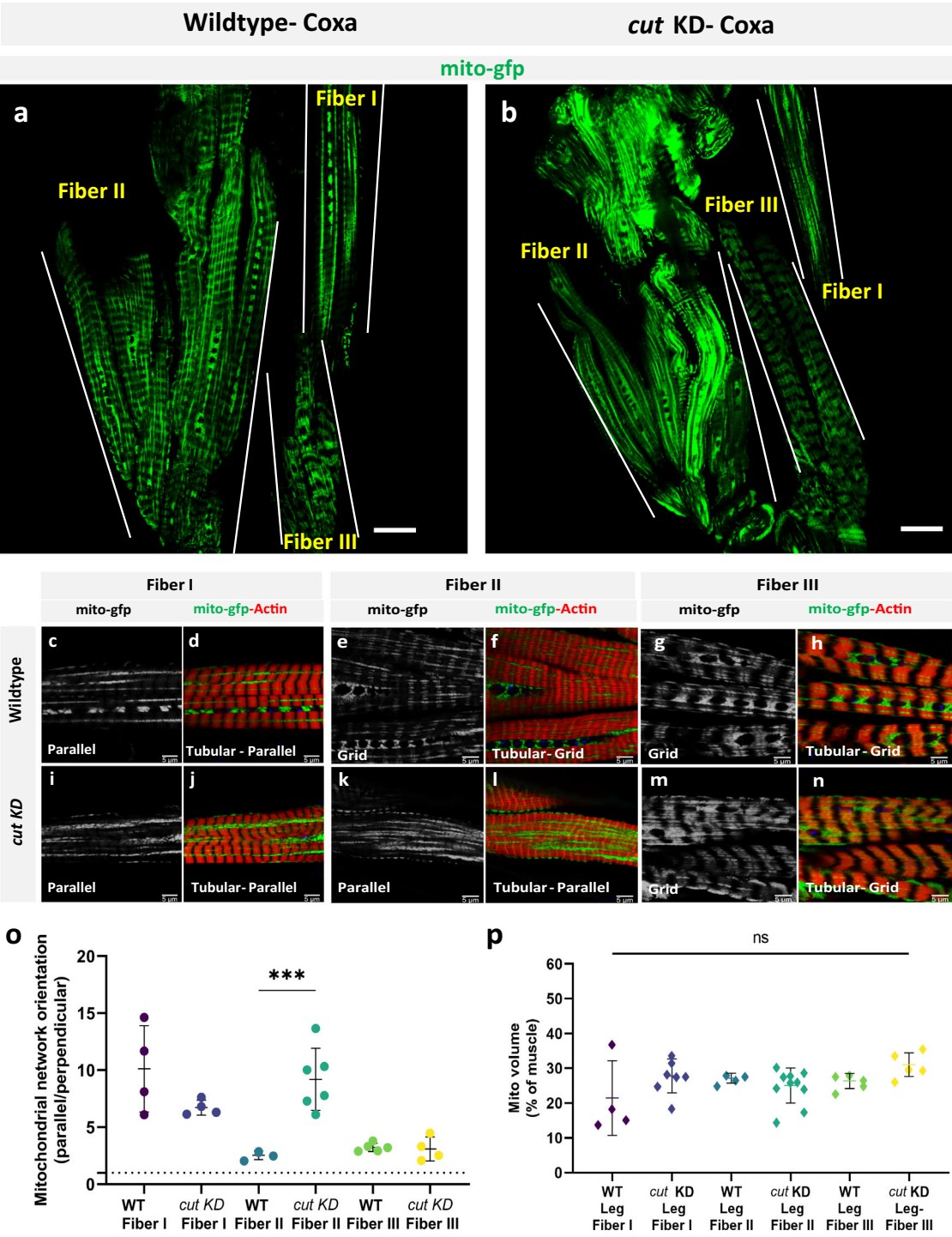

**Fig. 6 | *cut* regulates conversion of mitochondrial networks configuration, but not contractile type, in Fiber II of Drosophila leg muscles. a** Adult wildtype leg coxa muscles showing distinct parallel and grid-like mitochondrial networks (mito-gfp) in Fiber I, II, and III (demarcated by dotted lines). **b** *cut* KD converts grid-like mitochondrial networks to parallel in Fiber II (Scale Bars: 20 μm). **c, d** Wildtype leg Fiber I showing parallel mitochondrial networks (mito-GFP) and tubular muscle fiber (phTRITC). **e, f** Wildtype leg Fiber II and **g, h** Fiber III showing grid-like mitochondrial networks (mito-GFP) and tubular muscle fiber (phTRITC). **i, j** *cut* KD Fiber I showing parallel mitochondrial networks (mito-GFP) and tubular muscle fiber (phTRITC). **k, l** *cut* KD Fiber II showing parallel mitochondrial networks (mito-GFP), unlike wildtype Fiber II, while fibers remain tubular (phTRITC). **m, n** *cut* KD Fiber III showing grid-like mitochondrial networks (mito-GFP) and tubular muscle fibers

(phTRITC). (Scale Bars: 5 μm for all). **o** Quantification of mitochondrial network orientation. Dotted line represents parallel equal to perpendicular. *mito-gfp;mito-mcherry;mef2-Gal4* used as Wildtype, WT (WT-Fiber I, *n* = 4 animals; *cut* KD-Fiber I, *n* = 4 animals; WT-Fiber II, *n* = 3 animals; *cut* KD-Fiber II, *n* = 6 animals; WT-Fiber III, *n* = 4 animals; *cut* KD-Fiber III, *n* = 4 animals). **p** Mitochondrial volume as a percent of total muscle volume. *UAS-mito-gfp;UAS-mitoOMM-mcherry;mef2-Gal4* used as wildtype (WT-Leg Fiber I, *n* = 4 animals; *cut* KD-Leg Fiber I, *n* = 7 animals; WT-Leg Fiber II, *n* = 4 animals; *cut* KD-Leg Fiber II, *n* = 10 animals; WT-Leg Fiber III, *n* = 5 animals; *cut* KD-Leg Fiber III, *n* = 5 animals). Each point represents value for each animal dataset. Bars represent mean ± SD. Significance determined as *p* < 0.05 from one way ANOVA with Tukey's (*, *p* ≤ 0.05; **, *p* ≤ 0.01; ***, *p* ≤ 0.001; ****, *p* ≤ 0.0001; ns, non-significant).

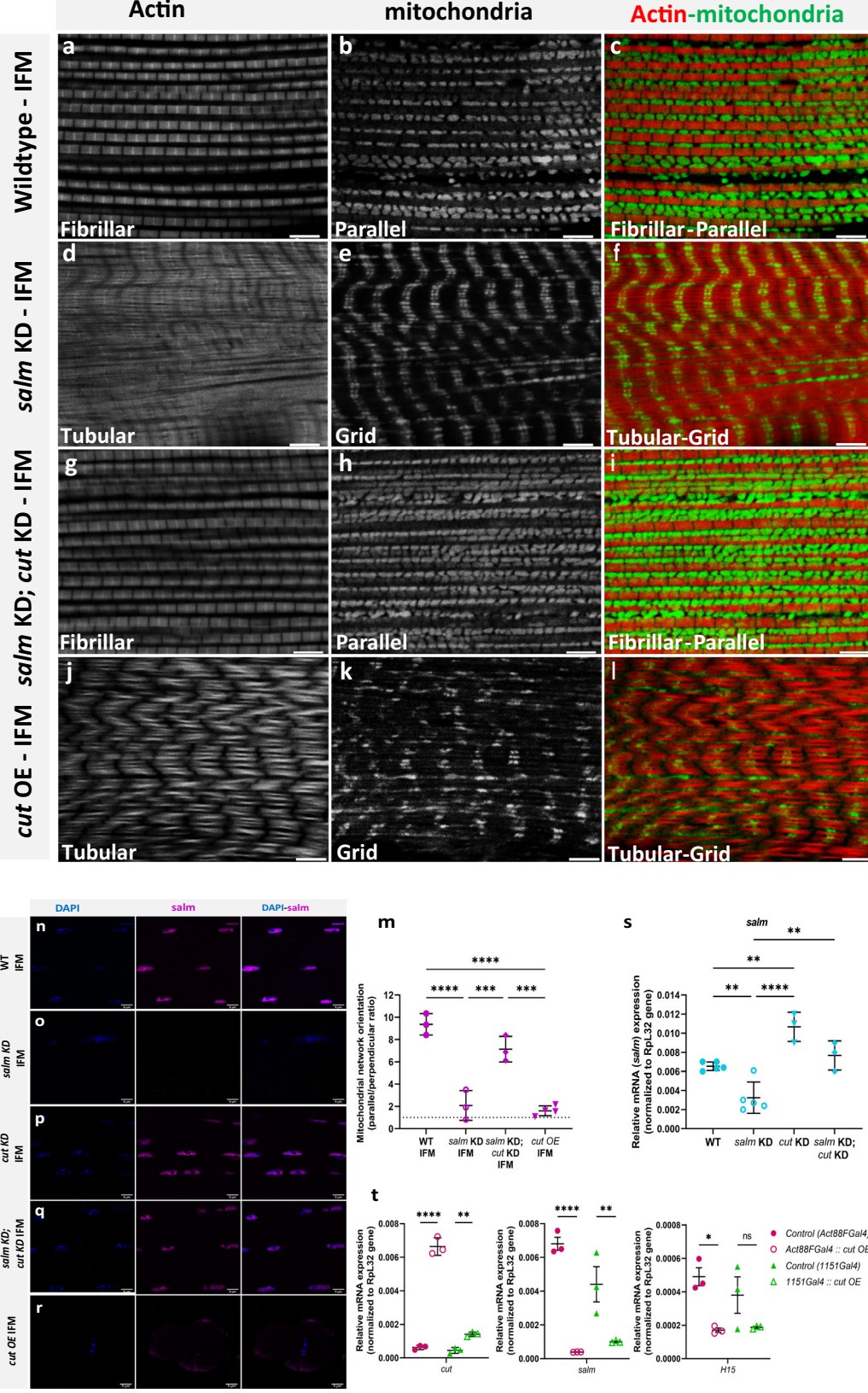

## Discussion

Muscle contraction is an energetically demanding process occurring in a tightly packed cellular environment, thereby requiring close coordination between the architectures of the contractile and metabolic machineries to optimally support the function the muscle cell. While regulation of metabolic and contractile properties occurs together

early in the muscle-type specification process[37,97,98], at which point(s) the regulation of mitochondrial network structure diverges into a pathway independent from contractile network structure is not well understood. Here, we utilized the naturally occurring functional differences among *Drosophila* muscles[61,63] combined with the known *Drosophila* muscle type specification factor *salm* to identify 142

**Fig. 7 | *cut* is a repressor of *salm* and *cut* KD rescues fiber-type switching and conversion of mitochondrial network organization in *salm* KD fibers.**
**a, b, c** Fibrillar flight muscles (IFMs) stained for F-actin (phTRITC) and mitochondria (MitoTracker) showing parallel aligned mitochondria between myofibrils.
**d, e, f** *salm* KD (*UAS-salm RNAi;UAS-mito-gfp;mef2*) shows fibrillar muscles switched to tubular muscle type and mitochondria (mito-gfp) converted to grid-like networks. **g, h, i** *salm KD; cut KD* shows fibrillar fiber type and parallel mitochondrial networks (MitoTracker) in IFMs similar to wildtype. **j, k, l** *cut*-OE shows tubular fiber type and grid-like mitochondrial networks (MitoTracker) in IFMs similar to *salm* KD (*UAS-salm RNAi;UAS-mito-gfp;mef2*) (Scale Bars: 5 μm for all). **m** Quantification of mitochondrial network orientation. Dotted line represents parallel equal to perpendicular (WT IFM, *n* = 3 animals; *salm* KD(*UAS-salm RNAi;UAS-mito-gfp;mef2*) IFM,

*n* = 3 animals; *salm KD; cut KD* IFM, *n* = 3 animals; *cut*-OE, *n* = 4 animals). **n** Wildtype fibrillar IFM stained for nuclei (DAPI), and salm antibody showing salm expression in the nuclei. **o** *salm* KD IFM showing decreased salm expression. **p** *cut* KD IFM and **q** *salm KD; cut KD* showing restored salm expression in the nuclei. **r** *cut*-OE showing absence of salm expression in the nuclei (Scale Bars: 5 μm for all). **s** Quantification of *salm* transcript levels (WT IFM, *n* = 5; *salm* KD (*UAS-salm RNAi;UAS-mito-gfp;mef2*) IFM, *n* = 5; *cut* KD, *n* = 3; *salm KD; cut KD* IFM, *n* = 3). **t** Quantification(qPCR) of *cut, salm, and H15* transcript levels. Each point represents value for each dataset. Bars represent mean ± SD. Significance determined as *p* < 0.05 from one way ANOVA with Tukey's (*, *p* ≤ 0.05; **, *p* ≤ 0.01; ***, *p* ≤ 0.001; ****, *p* ≤ 0.0001; ns, non-significant).

proteins consistently associated with either fibrillar or tubular contractile types, parallel or grid-like mitochondrial networks, or *salm* expression level. By performing phenotypic assessments of the contractile and mitochondrial networks in the flight, jump, and leg muscles of flies with knockdown or overexpression of five additional transcription factors identified by our screen, we demonstrate that mitochondrial network configuration and contractile type can be regulated independently through evolutionarily conserved transcription factors *cut*, *salm*, and *H15*. Moreover, we show that the regulatory role of each of these transcription factors for mitochondrial network specification can be variable among the flight, jump, and leg muscles and even within different regions of the leg muscles.

The parallel mitochondrial networks running between the fibrillar myofibrils of the flight muscles can be converted to grid-like mitochondrial networks reminiscent of the tubular leg muscles in the absence of *salm* as shown here (Fig. 2g) and elsewhere[50] while the current paper was under revision. Further, we find here that *salm* regulation of mitochondrial network configuration in flight muscles is mediated by downstream regulator *H15* as *H15* overexpression in the absence of *salm* is sufficient to restore the flight muscles back to their wild-type parallel mitochondrial network phenotype (Fig. 5n–p). However, loss of *H15* alone resulted in the conversion of the flight muscles to a jump muscle phenotype with parallel mitochondrial networks located within a tubular contractile network (Fig. 4d, e). Thus, conversion of the contractile network of the flight muscles from fibrillar to tubular can occur with (*salm* KD) or without (*H15* KD) conversion of the mitochondrial network highlighting the capacity for independent regulation of the two major structural components within the flight muscle cell.

We found that the mitochondrial networks in the tubular muscles can take either a parallel (jump/leg Fiber I) or grid-like (leg Fiber II/III) configuration (Fig. 1) again demonstrating the independent nature of the mitochondrial network and contractile type specification processes. Whereas the mitochondrial properties of the flight muscles have been widely studied[46,47,60,99], there has been relatively little investigation into how metabolism may vary across tubular muscle types. Similar to the differences in mitochondrial content we have shown here among tubular muscles (Fig. 1t), two different levels of mitochondrial enzyme activity within the leg muscles, each higher than the activity of the jump muscle, has been reported previously[65]. However, the flight muscle, jump muscle, and Fiber I of the leg all have parallel mitochondrial networks despite largely different mitochondrial contents (Fig. 1u) and enzyme activity levels[65] suggesting that mitochondrial network configuration may be specified differently than mitochondrial content. Indeed, we find that *H15* KD converts the parallel mitochondrial networks of the jump and leg Fiber I muscles to more grid-like networks (Fig. 4) and *cut* KD converts the grid-like mitochondrial networks in Fiber II of the leg to parallel (Fig. 6) each without altering mitochondrial content (Fig. 6j and Fig. S14b) or the tubular nature of the myofibrils. Thus, regulation of mitochondrial network configuration also appears to occur independently from mitochondrial content and enzyme activity in addition to contractile

type. A separation of the mitochondrial content and network configuration specification processes may also explain why loss of *spargel*, the *Drosophila* PGC-1α ortholog, does not alter the location of mitochondria between the myofibrils in flight muscles despite having a significant effect on individual mitochondrial structure and metabolism[15].

Mitochondrial dynamics proteins governing the capacity for mitochondrial fission, fusion, and motility have been shown to regulate the structure of individual mitochondria, mitochondrial network formation, and/or maintenance of mitochondrial quality control across a variety of cell types[20,22,74–77]. However, consistent with images from previous studies in the flight muscle, we found that loss of *Marf* (mfn1/2 ortholog)[46–50], *Drp1*[52], *Fis1*[53], and *Miro*[54] did not change the parallel configuration of mitochondrial networks in the fibrillar muscles despite each gene modulating individual mitochondrial size (Fig. S4). Moreover, overexpression of both *Drp1* and *Miro* has also been shown to alter mitochondrial size, but not network configuration in *Drosophila* flight muscles[50,55]. Thus, these data indicate that regulation of individual mitochondrial size by mitochondrial dynamics proteins can occur independently from specification of mitochondrial network configuration. It should be noted that *Mef2-Gal4* driven overexpression of *Marf* and expression of a dominant negative (DN) form of *Drp1* were both found recently to cause conversion of the flight muscles to a tubular contractile apparatus[50]. However, the effects of *Marf* OE and DN-*Drp1* on flight muscle mitochondrial networks appeared variable across the images provided, and neither individual mitochondrial size nor network orientation were quantified precluding direct comparisons. What is clear from the available *Marf* OE and DN-*Drp1* data[50] is that the flight muscle mitochondrial networks did not take a grid-like configuration as seen in the wild-type leg muscles or *salm* KD flight muscles and instead appeared to vary between the wild-type phenotypes of the tubular DFM and jump muscles, which both have parallel mitochondrial network configurations despite the large increase in mitochondrial size and content in the DFMs. Thus, these data are consistent with the capacity of *Drosophila* muscles to regulate both contractile type and mitochondrial size separately from mitochondria network configuration as shown here.

We propose that the specification of mitochondrial network configuration is an independent process within the overall cellular design of the muscle cell. Through assessment of the contractile and mitochondrial networks in the flight, jump, and leg muscles of flies expressing wild-type, increased, and/or decreased levels of nine genes associated with mitochondrial dynamics or muscle contractile type, we find that mitochondrial network configuration, mitochondrial content, individual mitochondrial size, and contractile type can each be modulated without affecting the other parameters thereby demonstrating the independent nature of each of these aspects of muscle cell design. As shown here, the pathway regulating muscle mitochondrial network configuration involves *cut* upstream from *salm* and *H15* downstream from *salm* (Fig. S29). However, additional factors are likely involved. Indeed, transcription factors *extradenticle* (*exd*), *homothorax* (*hth*), and *vestigial* (*vg*) are each known to specify muscle

cell fate upstream from *salm*[50,67,78,95]. Moreover, vertebrate orthologs of *exd* (Pbx1-4), *hth* (Meis1-4), *vg* (Vgll1-3), and *H15* (Tbx15) have all been shown to play a role in muscle fiber type specification[81,100–102] indicating that the specific regulation of mitochondrial network configuration as shown here is likely an evolutionarily conserved process, although the specific roles of *salm* (sall1-4) and *cut* (cux1–2) orthologs have yet to be determined in mammalian striated muscles.

## Methods

### *Drosophila* strains and genetics
Genetic crosses were performed on yeast corn medium at 25 °C unless mentioned. W[1118] were used as controls and respective genetic backgrounds. Both males and females were used and grouped together due an observed lack of sex differences in mitochondrial network configuration in wild type muscles. *Mef2- Gal4* (III) was used to drive muscle specific gene knockdown and over expression of respective genes. Tub-Gal80;[ts] Mef2 Gal4 used for ectopic expression of *salm* in muscles. UAS-mito-GFP (II chromosome BS# 8442) was used for mitochondrial network visualization. UAS- mito-mcherry (III) was used for visualization of the outer mitochondrial membrane. *UAS- H15 RNAi* trip lines were used for muscle specific knock down of *H15* gene. *UAS-mito-mcherry* (BS# 66533), *Mef2-Gal4* (BS# 27390), *Act88F-Gal4* (III, BS# 38461), *1151-Gal4*[103,104] (I, gift from Dr. Upendra Nongthomba) *UAS-H15 RNAi* (V28415), *UAS-Drp1RNAi (BS# 51483)*, *Marf RNAi* (BS# 55189), *UAS-salm* (Dr. Frank Schnorrer), *UAS-salm RNAi* (V101052), *UAS-Miro RNAi* (V106683), *UAS-Fis1 RNAi* (BS# 63027), *UAS-cut RNAi* (BS# 33967, #29625), *UAS-H15/nmr1* (Dr. Rolf Bodmer), *UAS- cut*[105] (gift from Dr. Yuh Nung Jan, UCSF, USA), *cut-OE* (TOE.GS00041) (BS# 67524), *UAS-Cas9.P2; Mef2-GAL4* /TM6B, (BS# 67075), *UAS-dCas9/cyO; Mef2-GAL4* (BS# 67041), *H15-OE* (BS# 78722)[106], UAS-RFP.KDEL (BS# 30910, # 30909) (Dr. Richa Rikhy), Zasp52MI02908-mCherry[107] (Dr. Frieder Schöck). *UAS-pros ORF* (F004799, FlyORF), *UAS- pros RNAi* (BS# 26745), *UAS-Lmpt.ORF.3xHA*. (F001889), *UAS-Lmpt RNAi* (v105170, 100716). All other stocks were requested or obtained from the VDRC (Vienna) Drosophila stock center, Bloomington (BS#) Drosophila stock center or FLYORF (Zurich) Drosophila stock center. All chromosomes and gene symbols are as mentioned in Flybase (http://flybase.org).

### Mitochondrial networks staining of flight (IFM), jump(TDT), and leg muscles
2–3-day-old adult *Drosophila* thoraces were dissected in 4% paraformaldehyde (PF, Sigma) with fine scissors, isolating each muscle type: IFMs, jump muscles (TDT) and leg muscles. IFMs, TDT, leg muscles were fixed in 4% PF for 2 h, 1.5 h, and 1.5 h, respectively using a rotor then washed in PBSTx (PBS + 0.3% TritonX100) thrice each for 15 min. Mitochondrial staining dyes (Mito Tracker Red (M22425, Thermofisher, USA) and Acridine Orange nonyl bromide (A7847, Sigma, USA) or *DMef2-Gal4* driven UAS-mito-GFP or UAS-mito OMM-mcherry were used to image mitochondrial networks. Actin staining was performed by incubating 2.5 μg/ml of Phalloidin in PBS (Sigma, 1 mg/ml stock of Phalloidin TRITC) at 25 °C for 40 min for IFMs, 20 min for jump muscles and 1 h for leg muscles at RT, respectively. Tissues were mounted on a glass slide with Prolong Glass Antifade Mountant with NucBlue stain (P36985, Thermofisher, USA) and images were captured with Zeiss 780 confocal microscope.

### Immunohistochemistry
Fly thoraces were dissected for IFMs, jump muscles (TDT) and leg muscles in 4% PF and were processed as mentioned Rai et al.[47]. Briefly, each muscle group was fixed in 4% PF; thoraces with IFMs for 2 h, jump muscles for 1.5 h, and leg muscles for 1–1.5 h at room temperature (25 °C, RT) on a rotor. Samples were washed three times with PBSTx (PBS + 0.03% Triton X-100) for 15 min and blocked for 2 h at RT or overnight at 4 °C using 2% BSA (Bovine Serum Albumin, Sigma). Samples were incubated with respective primary antibody (Ab) at 4 °C

overnight and later washed three times for 10 min with PBSTx and incubated for 2.5 h in respective secondary Ab at 25 °C or overnight at 4 °C. Samples were incubated for 40 min with Phalloidin TRITC (2.5 μg/ml) (P1951, Sigma, USA) to counter stain samples and mounted using Prolong Glass Antifade Mountant with NucBlue stain and incubated for 20 min. Images were acquired with Leica SP8 STED 3X/Confocal Microscope and ZEISS 780 confocal microscope and processed using ImageJ and ZEN software (version 3.2.0.115) respectively. Antibodies used for the staining: Rabbit anti-salm-1 (1:500, gift from Dr. Tiffany Cook[108],), Rabbit anti-nmr1 (H15) (1:200, gift from Dr. James B. Skeath[109],), Mouse anti-cut (1:20, 2B10, DHSB), Alexa Fluor 594-labeled Goat anti-Mouse IgG (1:500, Cat# A-11032) and Alexa-Fluor-488-labeled anti-rabbit IgG (1:500, Cat# A32731, Thermofisher, USA). The mean fluorescence intensity of antibody immunostaining was measured as reported previously[110] using ImageJ. Three-dimensional rendering of mitochondria networks was performed using IMARIS 9.7.0 (http://www.bitplane.com/imaris/imaris).

### Real-time quantitative PCR
Total RNA was extracted from thorax muscles of 1–2 day-old adult flies using the TRIzol™ Plus RNA Purification Kit (A33254) according to the manufacturer's instructions. DNA digestion was carried out using TURBO DNA-free™ Kit (AM1907) and first-strand synthesis was performed using 500 ng of total RNA using SuperScript™ VILO™ cDNA Synthesis Kit (Thermo Fisher Scientific, 11754050). Standard curves were generated using concentrated cDNA from all samples. Amplification was detected on the QuantStudio3 Real-Time PCR System (Thermo Fisher Scientific). RLP32 reference gene was used for normalization and reactions were performed in triplicates. The TaqMan Probes *salm* (Assay ID: Dm01804248_g1), *cut (Assay ID: Dm01837171_m1) H15* (Assay ID: Dm01804677_m1), *fln* (Assay ID:Dm01823176_g1), *Act88F* (Assay ID:Dm02362815_s1), *TpnC4* (Assay ID:Dm01815264_m1) and *RLP32* (Assay ID: Dm02151827_g1) were used.

### Behavioral assays
**Flight test**. The flight was assayed at room temperature (22 °C) using a flight box, as detailed earlier[46,47]. Two to three-day-old flies were independently scored for flight ability as up flighted (U), horizontally flighted (H), down flighted (D), or flightless (F). Each fly was tested thrice.

**Jump test**. Jump tests were performed for the 2–3 days old adult flies as described previously[111]. Adult fly wings were removed and flies were allowed to recover for 24 h at room temperature (RT). Then, the flies were placed on a platform raised above a sheet of white paper and encouraged to jump from the platform using a paintbrush. The point of landing on the paper was marked and the distances were measured from the edge of the platform to the marked point in mm. Each fly was tested thrice and the average jump distance was calculated.

**Climbing assay**. Two–three-day-old flies were divided into three groups of ten flies in each vail and allowed to recover. Climbing assay was performed as previously described[112]. Groups of ten flies were placed in an empty climbing vial and then tapped down to the bottom. They were allowed 18 s to climb up to a line marked 5 cm from the bottom of the vial. The number of flies above the 5 cm mark at 18 s was recorded as the percentage of flies able to climb.

### Image analysis and quantifications
**Individual mitochondrial area and aspect ratio**. Individual mitochondria were traced using the freehand tool of ImageJ software (https://imagej.net) on 2D light microscopic images. Individual mitochondrial area and aspect ratio (ratio of major axis/minor axis) were calculated using ImageJ software[46].

**Mitochondrial and ER content.** Light microscopic images were opened using ImageJ and binarized and the total mitochondrial volume was determined as the percentage of binarized mitochondrial pixels per total muscle fiber pixels. ER content was calculated as the percentage of binarized ER pixels per total muscle fiber pixels

**Mitochondrial network analysis.** Mitochondrial network analysis was performed as described using ImageJ software[31]. The mitochondrial network orientation analysis was performed on 2D images, where the mitochondrial image was rotated such that the horizontal axis was parallel to the direction of muscle contraction. The images were then binarized and the OrientationJ Distributions plugin was used to determine the angles of the mitochondrial network. Parallel mitochondria were determined as those with a ±0–10° angle to the axis of muscle contraction and perpendicular mitochondria were determined as those with a ±80–90° angle to the axis of muscle contraction.

## Proteomic analysis

**Drosophila muscle protein extraction.** Three to four days old *Drosophila* thoraces were chopped soon after freezing in liquid nitrogen and each muscle type dissected (IFMs(n100), Jump muscles (*n* = 200) -TDT and Leg muscles (*n* = 250-300), *salm* KD IFMs (*n* = 200) *salm* OE leg muscles (*n* = 25–300) in 70% alcohol. Muscle protein extraction preparation was followed as described in Kim et al., 2019. Muscle tissues were transferred to 90 μl of urea-based lysis buffer (6 M Urea, 2 M Thiourea, 50 mM Triethylammonium bicarbonate [TEAB]) at 1:5 ratio. Tissues were homogenized with ceramic beads using three steps of 45 s at 6500 rpm and 4 °C (Precellys® Cryolys Evolution, Bertin Technologies). Legs were sonicated for 30 s twice for better homogenization and muscle samples were collected.

Tissue lysates were further homogenized and centrifuged at 10,000 *g* for 10 min at 4 °C in microcentrifuge spin columns (QIAshredder, Qiagen) to obtain clear protein lysate. The extracted protein supernatants were transferred to 1.5 ml microtubes for further processing. Protein concentration was estimated by Bradford assay (ThermoFisher Scientific). Briefly, 100 μg of each sample was digested with trypsin, labeled with Tandem Mass Tag (TMT) 11plex labeling reagent kit following manufacturer's instructions (Thermo Fisher Scientific), quenched with 5% hydroxylamine, and combined to make a total of 1 mg in a single microcentrifuge tube. The combined samples were desalted using a 1 cc Oasis HLB cartridge (Waters) following manufacturer's instructions and speedvaced to dryness.

**Offline HPLC peptide fractionation.** High pH reversed-phase liquid chromatography was performed on an offline Agilent 1200 series HPLC. Approximately, 1 mg of desalted peptides were resuspended in 0.1 ml 10 mM triethyl ammonium bicarbonate with 2% (v/v) acetonitrile. Peptides were loaded onto an Xbridge $C_{18}$ HPLC column (Waters; 2.1 mm inner diameter × 100 mm, 5 μm particle size), and profiled with a linear gradient of 5–35% buffer B (90% acetonitrile, 10 mM triethyl ammonium bicarbonate[TEAB]) over 60 min, at a flowrate of 0.25 ml/min. The chromatographic performance was monitored by sampling the eluate with a diode array detector (1200 series HPLC, Agilent) scanning between wavelengths of 200 and 400 nm. Fractions were collected at 1 min intervals followed by fraction concatenation. Fifteen concatenated fractions were dried and resuspended in 0.01% formic acid, 2% acetonitrile. Approximately 500 ng of peptide mixture was loaded per liquid chromatography-mass spectrometry run.

**Mass spectrometry.** All fractions were analyzed on an Ultimate 3000-nLC coupled to an Orbitrap Fusion Lumos Tribrid instrument (Thermo Fisher Scientific) equipped with a nanoelectrospray source. Peptides were separated on an EASY-Spray $C_{18}$ column (75 μm × 50 cm inner diameter, 2 μm particle size and 100 Å pore size, Thermo Fisher Scientific). Peptide fractions were placed in an autosampler and

separation was achieved by 120 min gradient from 4 to 24% buffer B (100% ACN and 0.1% formic acid) at a flow rate of 300 nL/min. An electrospray voltage of 1.9 kV was applied to the eluent via the EASY-Spray column electrode. The Lumos was operated in positive ion data-dependent mode, using Synchronous Precursor Selection (SPS-MS3). Full scan MS1 was performed in the Orbitrap with a precursor selection range of 380–1500 m/z at nominal resolution of $1.2 \times 10^5$. The AGC target and maximum accumulation time settings were set to $4 \times 10^5$ and 50 ms, respectively. MS2 was triggered by selecting the most intense precursor ions above an intensity threshold of $5 \times 10^3$ for collision induced dissociation (CID)-MS2 fragmentation with an AGC target and maximum accumulation time settings of $2 \times 10^4$ and 75 ms, respectively. Mass filtering was performed by the quadrupole with 0.7 m/z transmission window, followed by CID fragmentation in the linear ion trap with 35% normalized collision energy in rapid scan mode and parallelizable time option was selected. SPS was applied to co-select ten fragment ions for HCD-MS3 analysis. SPS ions were all selected within the 400–1200 m/z range and were set to preclude selection of the precursor ion and TMTC ion series. The AGC target and maximum accumulation time were set to $1 \times 10^5$ and 150 ms (respectively) and parallelizable time option was selected. Co-selected precursors for SPS-MS3 underwent HCD fragmentation with 65% normalized collision energy and were analyzed in the Orbitrap with nominal resolution of $5 \times 10^4$. The number of SPS- MS3 spectra acquired between full scans was restricted to a duty cycle of 3 s.

**Data processing.** Raw data files were processed using Proteome Discoverer (v2.3, Thermo Fisher Scientific), with Mascot (v2.6.2, Matrix Science) search node. The data files were searched against Translated EMBL (TrEMBL) *Drosophila melanogaster* protein sequence database (uniport.org), with carbamidomethylation of cysteine, TMT 11-plex modification of lysines and peptide N-terminus set as static modifications; oxidation of methionine and deamidation of asparagine and glutamines as dynamic. For SPS-MS3, the precursor and fragment ion tolerances of 10 ppm and 0.5 Da were applied, respectively. Up to two-missed tryptic cleavages were permitted. Percolator algorithm was used to calculate the false discovery rate (FDR) of peptide spectrum matches, set to *q* value 0.05.TMT 11-plex quantification was also performed by Proteome Discoverer by calculating the sum of centroided ions within 20 ppm window around the expected m/z for each of the 11 TMT reporter ions. Spectra with at least 60% of SPS masses matching to the identified peptide are considered as quantifiable PSMs. Quantification was performed at the MS3 level where the median of all quantifiable PSMs for each protein group was used for protein ratios. Only proteins detected in all samples were included for analysis. A relative protein abundance threshold of greater than or equal to 2.0 or less than or equal to 0.5 was used to determine proteins which were differentially expressed between muscle types and proteins with abundances within 25% were considered of similar abundance.

**FIB-SEM imaging.** 2–3 days old flies were dissected on standard fixative solution (2.5% glutaraldehyde, 1% paraformaldehyde, and 0.12 M sodium cacodylate buffer) and processed for FIB-SEM imaging as described previously[31]. Muscles were then transferred to the fresh fixative Eppendorf tube and fixed overnight at 4 °C. Samples were washed in 0.1 M cacodylate buffer for 10 min, three times. Later, fixed tissues were immersed in a solution of 3% KFeCN in 0.2 M cacodylate buffer with 4% $OsO_4$ on ice for 1 hr. Three washes with distilled water for 10 min each were then performed following incubation in filtered TCH solution for 20 min at room temperature. Three washes with water for 10 min each, then incubated for 30 min in 2% $OsO_4$ on ice. Incubated samples were washed with distilled water for 10 min, three times each. Tissues were transferred to the freshly made 1% uranyl acetate at 4 °C overnight. Furthermore, samples were washed with distilled water three times for 10 min, then incubated with lead

aspartate solution at 60 °C for 30 min. Repeated washing step with warm distilled water and then dehydrated tissues with increasing order of alcohol percentage (20%, 50%, 70%, 90%, 95%, 100%, 100%) at room temperature. Later, samples were transferred to the freshly made 50% Epon resin in alcohol and incubated for 4–5 h at room temperature and then replaced 50% Epon by the 75% Epon incubated overnight at room temperature. Samples were transferred to the freshly prepared 100% Epon resin and incubated for 1 h, repeated the step, and finally transferred to the 100% Epon for 4 h. Tissues were transferred to the stub with as little resin as possible and tissues were incubated for polymerization on the stub for 48 h at 60 °C.

FIB-SEM images were acquired using a ZEISS Crossbeam 540 with ZEISS Atlas 5 software (Carl Zeiss Microscopy GmbH, Jena, Germany) and collected using an in-column energy selective backscatter with filtering grid to reject unwanted secondary electrons and backscatter electrons up to a voltage of 1.5 kV at the working distance of 5.01 mm. FIB milling was performed at 30 kV, 2–2.5 nA beam current, and 10 nm thickness. Image stacks within a volume were aligned using Atlas 5 software (Fibics Incorporated, Ontario, Canada) and exported as TIFF files for analysis. Voxel size was set at $10 \times 10 \times 10$ nm.

FIB-SEM volumes were segmented in semi-automated fashion using Ilastik[113] machine learning software as described previously[31]. Briefly, FIB-SEM image volumes were binned to 20 nm in ImageJ, saved as 8-bit HDF5 files, and imported into Ilastik. Pixel classification training using all available features was performed for the mitochondria, sarcoplasmic reticulum+t-tubules and all other pixels and exported as 8-bit probability files. The resultant HDF5 files were imported back into ImageJ and binarized using a 50% threshold. The binary structures were filtered using the Remove Outliers plugin in ImageJ using a 3-10 pixel radius and 1.5-2 standard deviations and then rendered in Imaris (Bitplane).

**Statistics and reproducibility.** To verify the reproducibility of the original phenotype, we ran a minimum of two independent experiments, which resulted in similar results. Every sample quantified relates to an individual animal, so all samples are biological replicates. Detailed information on the number of samples or animals used for each quantification is shown in the figure legends.

Quantitative data were analyzed in Excel 2016 (Microsoft) and statistical tests were performed using Prism 9 (GraphPad). All comparisons of means between Fiber types (IFMs, TDTs, Leg Fiber I, Fiber II, and Fiber III) and between gene knockdown and overexpression groups were performed using one-way analysis of variance with Tukey's HSD (honestly significant difference) post hoc test. Differences in Cut antibody fluorescence between cut KD and WT were evaluated by comparing the means from each dataset using unpaired $t$ test with Welch's correction. A $p$ value < 0.05 was used to determine statistical significance. Source data are provided as a Source Data file.

### Reporting summary
Further information on research design is available in the Nature Portfolio Reporting Summary linked to this article.

## Data availability
Protein abundance data for the proteomics screen has been provided as Supplemental Dataset 1. Raw mass spectrometry data have been uploaded to MassIVE[114] repository under accession number MSV000088173. Data can also be accessed through ProteomeXchange under accession number PXD028878 Source data for all figures are provided with this manuscript. All raw images used in this work are available upon reasonable request. Source data are provided in this paper.

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

## Acknowledgements

We thank the NHLBI Light Microscopy Core for providing confocal microscope access. We thank Dr. Hong Xu (NHLBI) for fruitful discussions as well as for sharing fly lines and reagents, and Dr. Upendra Nongthomba (Indian Institute of Science), Dr. Richard Cripps (University of New Mexico), and Dr. Frank Schnorrer (Institute for Developmental Biology) for sharing fly lines with us. We acknowledge the Vienna *Drosophila* Resource Center and the Bloomington Stock Center for providing us with fly stocks and reagents. This work was supported by the Division of Intramural Research of the National Heart Lung and Blood Institute and the Intramural Research Program of the National Institute of Arthritis and Musculoskeletal and Skin Diseases.

## Author contributions

P.K. performed all dissections and the generation and maintenance of fly lines. P.K. and B.G. designed, and P.K. performed the confocal imaging and P.K. and A.A. performed the proteomic experiments. P.K. performed all confocal image analyses, and P.K., A.A., and B.G. performed proteomic analyses. C.K.E.B. collected and P.T.A. and B.G. processed and rendered the FIB-SEM datasets. P.K. and B.G. wrote and P.K., P.T.A., A.A., C.K.E.B., and B.G. edited the manuscript.

## Competing interests

The authors declare no competing interests.
