## [Peer Review File · Nature Communications]

Identification of Evolutionarily Conserved Regulators of Muscle Mitochondrial Network OrganizationReviewers' comments:

Reviewer #1 (Remarks to the Author):

Manuscript by Katti and Glancy demonstrates that the morphology of muscle myofibrils and mitochondrial network is different in flight, jump, and walking muscles of *Drosophila melanogaster*. It also shows that knocking down of Spalt major (salm) or H15 affects mitochondrial organisation and converts the fibrillar flight muscles to tubular.

However, none of these findings are unexpected. It has been previously demonstrated that master transcriptional regulators such as *Drosophila* Spalt major are sufficient to convert one muscle type into another and recent evidence suggest also that SALL4 (a mammalian ortholog of salm) affects mitochondrial biogenesis (PMID: 31446059 showed that SALL4 binds approximately 50% of mitochondrial genes). Also, the Tbx15 (a mammalian ortholog of H15) has been shown to regulate fibre identity and muscle metabolism.

The manuscript is also very descriptive and mostly limited to images and there is no numerical data (mitochondrial size, density or mitochondrial network organisation) to support the observations and conclusions.

There are also no experiments to support the conclusion that H15 regulates mitochondrial network organisation and muscle contractile type downstream of salm. Such a conclusion requires experiment where H15 is overexpressed in salm KD model.

Thus, although the manuscript is interesting it has too many limitations and I cannot support its publication in Nature Communications.

--

Reviewer #2 (Remarks to the Author):

The aim of the present study is to understand organization of mitochondrial networks in different types of muscles in *Drosophila*. They use classic fly genetics, microscopy and TMT-labeling proteomics to identify novel factors that may orchestrate mitochondrial networks.

The study is technically well performed and described in the manuscript, yet there are some concerns that should be addressed before it can be reconsidered for publication.

1. What is the reproducibility of the proteomics analyses? What is the N of the individual muscle types? data from individual muscle types should be plotted in a Venn diagram and at least provided in the suppl data.
2. The proteomics data should be (re)examined using some kind of clustering method to identify clusters of regulated proteins. Such a figure should at least be provided as a suppl figure. This should be complemented by some kind of functional analysis of protein functions e.g. GO term enrichment analysis.
3. What is the dynamic range of detection? how many orders of magnitude?
4. Are the different muscle types homogenized evenly? How is the extend of lysis of the different muscle type checked?

Minor:

The authors state that they have digested 100 ug protein with trypsin, but resuspended 1 mg of

desalted peptides in TEAB. How is this possible? please explain or rephrase in the text?

The authors state that samples are further homogenized by microcentrifuge spin columns. Such columns do not further homogenize samples, they simply just clear the homogenates.

The proteomics data should be uploaded to a commonly available repository/database.

--

Reviewer #3 (Remarks to the Author):

Summary: The manuscript of Katti and Glancy details their investigation of mitochondrial sizes, shapes, and organization in different muscles of adult *Drosophila*. This study follows from a descriptive paper published by the Glancy lab, where mitochondrial organization was characterized and quantified in different mammalian muscle fiber types. By turning to the *Drosophila* system, the authors could more readily identify key genes responsible for different mitochondrial morphological characteristics and link them to muscle function. This manuscript begins with a description of mitochondrial morphology in the flight, jump, and leg muscles. The authors then use genetic approaches to determine whether known genes that are responsible for regulation of different mitochondrial behaviors (fusion, fission, transport). These appear not to affect mitochondrial organization. Subsequently, the authors focus on Spalt major (Salm), a transcription factor, previously identified by the Schnorrer group as important for muscle fiber type specification (flight-fibrillar vs jump-tubular). The authors report that loss of *salm* in the flight muscle leads to a tubular myofibril organization and grid-like mitochondrial organization characteristic of some of the leg muscles. By contrast, the authors report that overexpression of *salm* in the leg muscles leads to a fibrillar-like myofibril organization and parallel organization of the mitochondria. The authors go on to do a proteomics study on the different adult muscle groups in control, *salm* loss and gain of function. These experiments focus their attention on H15, a member of the Tbox family of transcriptional regulators. They report that H15 can convert a parallel organization to a grid like pattern in some muscle. The authors conclude that mitochondrial organization can be regulated independent muscle contractile type.

Critique: The authors investigate a very interesting problem in cell biology: how are the mitochondrial networks and, in particular, how are mitochondrial sizes, shapes and organization built and maintained? This is a particularly important issue in muscle cells, which are known to have different energetic demands. Moreover, it is unknown if/how mitochondrial organization is linked to myofibril type and organization and what the factors that control mitochondrial organization are in these different muscle types. The premise of the paper, no doubt, is exciting.

However, the approach and the data fall very short of providing any insight to this area and don't support the conclusions of this paper.

Among the issues that need to be addressed:

1- Quantification of all phenotypes discussed and shown. The Glancy lab published a very nice paper in which they quantified mitochondrial shapes, sizes, and organization in mouse muscle fibers. This should be done here to characterize the sizes, shapes and the organization. Measurements and statistical data has to be included. In addition, the authors have not described how they assessed flight, jumping, or walking and as with the experiments with mitochondrial morphology, no quantification/statistics shown. This is important as if the mitochondrial pattern is truly affected, a connection to any impact on flight, jumping or walking would be exciting.

2- Better clarification of experimental design, methodology, and data.

A. Mitochondrial morphology in *Drosophila* is altered depending on dissection buffers (Wang et al., Mitochondrion 2016 from the Geisbrecht lab). What was used here?

B. Muscle phenotypes: i. How many adults were analyzed, how many different muscle fibers? How old were the flies? Are both sexes represented? From where in the flight muscles, jump muscles or leg muscles are the images? How many images? How many type 1, 2, and 3 leg muscles? Why is there no TEM to validate parallel vs grid? Nothing is mentioned here.

ii. Have the authors considered the possibility that the type II (page 5, line 133) and type III fibers in the walking muscles of the coxa are the same? Based on the images provided in Figure 1k and Supp. Figure 1c, the sarcomeres stained for F-actin show no discernible H zones, actin-free regions of the sarcomere, as seen in Figures 1a, 1d and 1i. The lack of a H-zone suggests that the muscles are contracted and hence appear different from the Type III fibers, making the type II fibers an artifact. Experiments to address this should be conducted.

C. Experimental design I: The authors use mito-gfp in some situations and mito-mcherry in others. Are they the same? Also, it is accepted practice in the *Drosophila* field that controls mimic the experimental situation, so Mef2GAL4 X UAS-gfp (for example) as the control for Mef2GAL4X UAS-mito-gfp. Also the authors need to control for GAL4 dilution effect, as it isn't acceptable to compare UAS-mito-gfp to UAS-mito-gfp, UAS-Gene of interest. The control needs to have UAS-mito-gfp plus another UAS. This is important as the authors may not be getting as much knockdown or overexpression in these conditions, as they expect. QRT PCR, Westerns, or IF with quantification should be done to assess the knockdown and overexpression manipulations.

D. The authors show data suggesting that there are different mitochondrial networks between muscle types. Within the tubular muscles (jump and leg) there are parallel (jump, type 1 leg) and grid (type 2, 3). However in subsequent text and most figures, there is "Leg-coxa" and the reader is unclear whether it is fiber type 2 or 3. The authors say "walking muscles" in the text and in some figures, but they can be precise and tell us which, and with measurements tell us the amount of transformation from one type of organization (be it mitochondrial networks or myofibril organization). Hence it is difficult to assess the data and the conclusions drawn.

E. Proteomics. The proteomics analysis outlined in Figure 4 is not clearly explained. Validation of the proteomics study should be included. Western analysis can be done to show that known targets of Salm are appropriately regulated as suggested. In addition the authors conclude, in Line 239, that "These results are consistent with salm as a negative regulator of most of its targets to maintain the fibrillar nature of the flight muscles in *Drosophila*." How does this fit with the authors own genetic data (expression of Salm in some of the tubular muscle – which normally doesn't express Salm-- converts those muscles to fibrillar? Moreover it doesn't fit with published data from the Schnorrer group. This needs clarification.

D. H15 knockdown. As these experiments are important for the author's major conclusions, the amount of kd must be quantified in the muscle types. The authors conclude from their data that H15 is necessary for the grid like arrays in some leg muscles. The data shown, however, are not convincing. It looks like there are fewer mitochondria and as a result, the parallel arrays are not readily observable and the authors interpret the pattern as a grid array. Again, TEM data would support the point that the authors wish to make. Not clear why overexpression studies were not included to make a point of sufficiency.

E. Figure prep: All figures can be significantly improved. When showing individual channels, greyscale images should be used for clarity. Better images should be possible. High magnification insets have been provided for some panels. These insets should be provided for all the panels along with scale bars. Similarly, deciphering scale bar size from the images is nearly impossible and should be mentioned in the Figure legends. This is true for all scale bars and high-magnification insets throughout the paper.

3. Writing – the authors should carefully re-read the manuscript as there are issues of clarity throughout. The introduction and discussion are long-winded, and not arranged logically. There are numerous grammatical mistakes throughout the paper. Moreover they should carefully consider their word choice, as the data shown do not show what they conclude. Listed below are some of the concerns regarding the language and grammar used in the manuscript. Some examples:

The abstract is drastically over the word limit at 248 words.

Mitochondria is plural. Individual mitochondria are called mitochondrion.

please italicize *Drosophila* whenever used in the paper.

The nomenclature in *Drosophila* is: Salm: protein and *salm*: gene.

Line 43: Misexpression of ...spalt major (*salm*) converted the fibrillar flight muscles to tubular.... Loss of *salm* converts fibrillar to tubular. Gain of *salm* converts tubular to fibrillar. The sentence is not clear.

Line 80. Myogenic factors, like myostatin, MyoD, Mef2, and calcineurin These are all different types of proteins, including signals(myostatin) and transcriptional regulators (MyoD and Mef2). Best to revise this sentence to clarify for a general audience.

Line 98 neurogenic?

Line 161- "flighted" is not a valid word to describe the flight capacity of flies.

Line 293: different mitochondrial networks in *Drosophila* muscles are reflective of functional properties of the muscle fibers that they reside in. This is a bit of an overinterpretation as no functional aspects were shown here.

Line 302. 'Walking muscles are the most frequently used muscle group " data? Reference?

Line 305: "...walking muscles are comprised of heterogenous types of muscle fibers..." The authors haven't shown in this manuscript that the leg muscles (figure 1) are composed of different fiber types. Moreover, are the authors suggesting that each "group" (leg muscles 1, 2, 3) is made up of different myofiber types like one see in the TA muscle of mammals?

Line 321. Since there is no quantification, the authors really can't draw this conclusion. Later in this para, "abnormal individual mitochondrial morphology". Again quantification would support this. If done.

Line 335, need to introduce, *exd*, *hth*, *vg* ..these symbols don't mean a lot to the general reader.

Reviewer #1 (Remarks to the Author):

We thank reviewer #1 for the comments which have helped to significantly strengthen the paper.

Manuscript by Katti and Glancy demonstrates that the morphology of muscle myofibrils and mitochondrial network is different in flight, jump, and walking muscles of *Drosophila melanogaster*. It also shows that knocking down of Spalt major (*salm*) or H15 affects mitochondrial organization and converts the fibrillar flight muscles to tubular.

However, none of these findings are unexpected. It has been previously demonstrated that master transcriptional regulators such as *Drosophila* Spalt major are sufficient to convert one muscle type into another and recent evidence suggest also that SALL4 (a mammalian ortholog of *salm*) affects mitochondrial biogenesis (PMID: 31446059 showed that SALL4 binds approximately 50% of mitochondrial genes).

Response: We thank the reviewer for making these points. As shown by the reviewer, *salm* is well known as fiber-type specifying gene in *Drosophila* muscle, and this is why we chose to use misexpression of this gene for our screen to identify potential new regulatory factors of mitochondrial network configuration. However, we would like to point out that not all master transcriptional regulators involved in muscle metabolism result in changes in mitochondrial network configuration. For example, loss of the master regulator of mitochondrial biogenesis (*spargel*, the PGC1a ortholog) in *Drosophila* flight muscle did not result in conversion of the contractile apparatus or the location of mitochondria between the myofibrils despite having a significant effect on individual mitochondrial structure and metabolism (Ng et al. *Neurobiology of Aging*, 2017). We have reproduced the lack of conversion of contractile or mitochondrial networks in flight muscles and extended them to the jump and leg muscles including a figure below for the reviewers examination (Images from RNAi line BS# 57043 crossed with *Mef2-Gal4*). Thus, it should not be expected that transcription factors known to modulate mitochondrial metabolism or biogenesis will alter mitochondrial network configuration. We have now added this point to the main text. Moreover, we have now quantified mitochondrial content in addition to mitochondrial network configuration which has allowed us to demonstrate that mitochondrial network configuration changes can occur without altering mitochondrial content and vice versa. Thus, this provides further evidence that regulation of mitochondrial biogenesis (i.e. mitochondrial content) is a separate process from regulation of mitochondrial network structure which in and of itself is a novel finding for the mitochondrial field that can be attributed directly to the reviewers' helpful comments regarding the previous lack of quantitative data below.

Figure for Reviewers: Spargel (PGC1a ortholog) KD does not alter mitochondrial network configuration or contractile type in *Drosophila* flight, jump, or leg muscles.

Also, the Tbx15 (a mammalian ortholog of H15) has been shown to regulate fibre identity and muscle metabolism.

Response: A major goal here was not to just identify factors related to fiber identity, but to specify factors that could regulate mitochondrial network type separately from contractile type. Loss of H15 resulted in a change in contractile type in flight muscles, but not mitochondrial network type. However,

loss of H15 in jump or leg muscles changed the mitochondrial network but not the contractile type. Thus, while an ortholog of H15 has been shown to regulate fiber type, there was no indication from the previous work that H15 would regulate mitochondrial networks and contractile type differently in a muscle type-specific manner. Moreover, there was previously no information on how H15 worked with other transcription factors such as salm to induce fiber type changes in muscle. Thus, we have provided novel information that H15 operates downstream of salm which has been strengthened thanks to the reviewer's rescue experiment suggestion below. Also, we have now examined the role of another transcription factor, prospero, whose mammalian ortholog (prox1) is well known to alter fiber type (e.g. Kivelä et al. Nature Communications, 2016). We show that while prospero is critical to muscle function based on the reduced flight capacity in both muscle-specific knockdown and overexpression flies, misexpression of prospero did not alter mitochondrial network configuration in the flight, jump, or leg muscles again indicating that mitochondrial network configuration may be controlled separately from known regulators of muscle contractile or metabolic properties.

To specifically address the reviewers concern about novelty, we investigated the role of three additional transcription factors (including prospero) in the regulation of mitochondrial network configuration. We now show that transcription factor cut can also regulate mitochondrial network configuration in leg muscles. While cut has been shown to be important to flight muscle development in *Drosophila*, we are unaware of any information linking cut or its mammalian orthologs (Cux1-2) to the regulation of mitochondrial structure, function, or of non-flight muscles. Additionally, we demonstrate that cut operates as an upstream repressor of salm as cut KD combined with salm KD rescues the wild type flight muscle phenotype by elevating salm back to wild type levels.

The manuscript is also very descriptive and mostly limited to images and there is no numerical data (mitochondrial size, density or mitochondrial network organization) to support the observations and conclusions.

Response: We thank the reviewer for the comment. In this revised version, we have now included quantifications of individual mitochondrial area, cellular mitochondrial volume, and mitochondrial network orientation for the wild type muscles as well as the knockdown and/or overexpression of nine different genes. Thanks to these new data, we are now able to demonstrate that mitochondrial network configuration can be regulated independently from mitochondrial content and individual mitochondrial size in addition to contractile type. We believe this adds a major novel finding to this work that where mitochondria are placed within a cell (i.e. network configuration) is regulated separately from how much

mitochondria to put in a cell (i.e. mito content regulated by mito biogenesis factors) and how big to make each mitochondrion (regulated by mito dynamics factors).

There are also no experiments to support the conclusion that H15 regulates mitochondrial network organization and muscle contractile type downstream of *salm*. Such a conclusion requires experiment where H15 is overexpressed in *salm* KD model.

Response: We thank the reviewer for this comment. We previously performed immunohistochemical staining for H15 under *salm* KD condition and showed that H15 is down regulated in *salm* KD. However, *Salm* expression was unaltered following H15 KD. We have also now performed rescue experiments as suggested where H15 is overexpressed together with *salm* KD resulting in fibrillar myofibrils and parallel mitochondrial networks in 38% of the indirect flight muscles despite the continued lack of *salm*. Together, these data indicate that H15 operates downstream of *salm*.

Thus, although the manuscript is interesting it has too many limitations and I cannot support its publication in Nature Communications.

Response: We hope that the improvements in the manuscript guided by the reviewers comments, namely the addition of another regulator of mitochondrial network configuration, quantification of all discussed mitochondrial parameters which allowed for demonstration of mitochondrial network configuration as a separate entity from mitochondrial content and size, and direct demonstration of H15 being downstream of *salm* are sufficient to overcome the previous limitations and ease the reviewers concerns regarding this work.

Reviewer #2 (Remarks to the Author):

The aim of the present study is to understand organization of mitochondrial networks in different types of muscles in *Drosophila*. They use classic fly genetics, microscopy and TMT-labeling proteomics to identify novel factors that may orchestrate mitochondrial networks.

The study is technically well performed and described in the manuscript, yet there are some concerns that should be addressed before it can be reconsidered for publication.

We thank reviewer #2 for the comments which have helped to significantly strengthen the paper

1. What is the reproducibility of the proteomics analyses? What is the N of the individual muscle types? data from individual muscle types should be plotted in a Venn diagram and at least provided in the suppl data.

Response: We thank the reviewer for the comment. Additional details regarding the proteomic analyses have been included in the methods and supplemental figures section. With respect to the sample numbers, N= 2 replicates per pooled muscle types were used with each pool comprised of 100-300 whole muscle groups (i.e. one IFM is 12 DLM fibers, one TDT muscle group is 32 muscle fibers, and one leg is the whole leg). We created a Venn diagram and it did not distinguish the muscle types based on proteins found in one type as compared to another type. We created a PCA plot and Heatmap plot showing the reproducibility of the differences in muscle types and the data is included in the supplementary figures. We have also included a “Sample Abundances” chart displayed as a box-and-whisker plot. The plot displays the abundance of the peak intensity values for all proteins in the analysis. Altogether, these analyses indicate that there are clear and reproducible differences among the muscle types used here.

2. The proteomics data should be (re)examined using some kind of clustering method to identify clusters of regulated proteins. Such a figure should at least be provided as a suppl figure. This should be complemented by some kind of functional analysis of protein functions e.g. GO term enrichment analysis.

Response: As mentioned above, we have reanalyzed the proteomics data and the PCA plot and heatmap have been included in the supplementary figures. Further, Gene enrichment analysis was also performed and the data is now included in the supplementary figures now highlighting that metabolic and contractile proteins were the major differences observed across muscle types.

3. What is the dynamic range of detection? how many orders of magnitude?

Response: We thank the reviewer for this question. The now included “Sample Abundances” box and whisker plot (Fig. S11) shows that the peak intensity values for all proteins range across six orders of magnitude.

4. Are the different muscle types homogenized evenly? How is the extend of lysis of the different muscle type checked?

Response: We performed SDS gel electrophoresis of the lysed samples to check that the extent of lysis and homogenization was similar across samples.

Minor:

The authors state that they have digested 100 ug protein with trypsin, but resuspended 1 mg of desalted peptides in TEAB. How is this possible? please explain or rephrase in the text?

Response: We apologize for the lack of clarity. The protocol was clarified in the methods sections as follows

“Protein concentration was estimated by Bradford assay (ThermoFisher Scientific). Briefly, 100 ug of each sample was digested with trypsin, labeled with Tandem Mass Tag (TMT) 11plex labeling reagent kit following manufacturer’s instructions (Thermo Fisher Scientific), quenched with 5% hydroxylamine, and combined to make a total of 1 mg in a single microcentrifuge tube. The combined samples were desalted using a 1 cc Oasis HLB cartridge (Waters) following manufacturer’s instructions and speedvaced to dryness.”

The authors state that samples are further homogenized by microcentrifuge spin columns. Such columns do not further homogenize samples, they simply just clear the homogenates.

Response: We apologize for the lack of clarity. The protocol was clarified in the methods sections as follows

“Tissue lysates were further homogenized and centrifuged at 10,000 g for 10 min at 4°C in microcentrifuge spin columns (QIAshredder, Qiagen) to obtain clear protein lysate .”

The proteomics data should be uploaded to a commonly available repository/database.

We thank the reviewer for this suggestion. The proteomics data has now been uploaded to MassIVE (<http://massive.ucsd.edu/ProteoSAFe/QueryPXD?id=PXD028878>).

Reviewer #3 (Remarks to the Author):

Summary: The manuscript of Katti and Glancy details their investigation of mitochondrial sizes, shapes, and organization in different muscles of adult *Drosophila*. This study follows from a descriptive paper published by the Glancy lab, where mitochondrial organization was characterized and quantified in different mammalian muscle fiber types. By turning to the *Drosophila* system, the authors could more readily identify key genes responsible for different mitochondrial morphological characteristics and link them to muscle function. This manuscript begins with a description of mitochondrial morphology in the flight, jump, and leg muscles. The authors then use genetic approaches to determine whether known genes that are responsible for regulation of different mitochondrial behaviors (fusion, fission, transport). These appear not to affect mitochondrial organization. Subsequently, the authors focus on Spalt major (Salm), a transcription factor, previously identified

by the Schnorrer group as important for muscle fiber type specification (flight-fibrillar vs jump-tubular). The authors report that loss of *salm* in the flight muscle leads to a tubular myofibril organization and grid-like mitochondrial organization characteristic of some of the leg muscles. By contrast, the authors report that overexpression of *salm* in the leg muscles leads to a fibrillar-like myofibril organization and parallel organization of the mitochondria. The authors go on to do a proteomics study on the different adult muscle groups in control, *salm* loss and gain of function. These experiments focus their attention on H15, a member of the Tbox family of transcriptional regulators. They report that H15 can convert a parallel organization to a grid like pattern in some muscle. The authors conclude that mitochondrial organization can be regulated independent muscle contractile type.

Critique: The authors investigate a very interesting problem in cell biology: how are the mitochondrial networks and, in particular, how are mitochondrial sizes, shapes and organization built and maintained? This is a particularly important issue in muscle cells, which are known to have different energetic

demands. Moreover, it is unknown if/how mitochondrial organization is linked to myofibril type and organization and what the factors that control mitochondrial organization are in these different muscle types. The premise of the paper, no doubt, is exciting.

We thank reviewer for highlighting the importance and relevance of mitochondrial network regulation in cell biology in general and specifically in muscle cells. We hope that with the additional data and text provided that we have added new insights to this area, particularly that mitochondrial network configuration can be regulated separately from not only contractile type but also from mitochondrial content and individual mitochondrial size.

However, the approach and the data fall very short of providing any insight to this area and don't support the conclusions of this paper.

We thank reviewer #3 for constructive comments which have helped lead to a substantially improved paper. We hope that the addition of quantitative data on mitochondrial morphology throughout has now sufficiently supported our conclusions regarding the independent nature of mitochondrial network regulation. Specifically, we have now quantified mitochondrial network orientation in 48 different muscle types across the five muscle types compared here with misexpression (KD and/or OE) of nine different genes.

Among the issues that need to be addressed:

1- Quantification of all phenotypes discussed and shown. The Glancy lab published a very nice paper in which they quantified mitochondrial shapes, sizes, and organization in mouse muscle fibers. This should be done here to characterize the sizes, shapes and the organization. Measurements and statistical data has to be included. In addition, the authors have not described how they assessed flight, jumping, or walking and as with the experiments with mitochondrial morphology, no quantification/statistics shown. This is important as if the mitochondrial pattern is truly affected, a connection to any impact on flight, jumping or walking would be exciting.

Response: We thank the reviewer for the comment. In this revised version, we have included quantifications of mitochondrial area, mitochondrial volume, and mitochondrial network orientation for all conditions. We have also performed and described assays to evaluate flight, walking and jumping ability based on previous publications. All quantifications and statistical analyses are now included and support the conclusions in the revised manuscript.

2- Better clarification of experimental design, methodology, and data.

A. Mitochondrial morphology in *Drosophila* is altered depending on dissection buffers (Wang et al., Mitochondrion 2016 from the Geisbrecht lab). What was used here?

Response: We thank the reviewer for this comment and have now provided clarification of the experimental design and methodology including the details of the buffers used in the methods section. Specifically, the mentioned paper by Wang et al. compares mitochondrial morphology in larval muscles which were not assessed here, and they showed that both heat killing and relaxing buffer alter morphology when using antibody stains. We did not use either heat killing or relaxing buffer in our mitochondrial studies which were primarily performed using genetically targeted fluorophores. Additionally, we have now provided EM validation of our mitochondrial network phenotypes thanks to the reviewers suggestion below which further suggests that our dissections buffers did not grossly alter mitochondrial morphology in our light microscopy studies.

B. Muscle phenotypes: i. How many adults were analyzed, how many different muscle fibers? How old were the flies? Are both sexes represented? From where in the flight muscles, jump muscles or leg muscles are the images? How many images? How many type 1, 2, and 3 leg muscles? Why is there no TEM to validate parallel vs grid? Nothing is mentioned here.

Response: We thank the reviewer for this comment and have now clarified and included all relevant information in the revised manuscript including the N values per muscle in the figure legends as well as individual points in the quantitative figures. Also as now described in the methods, flies were 1-3 days old for all assays. Additionally, we have also added representative electron microscopy images of the indirect flight, jump, and grid-like leg muscles as suggested by the reviewer, but instead of TEM we used volume electron microscopy to demonstrate the full 3D nature of the mitochondrial networks in the wild type muscles as validation for the light microscopy images.

ii. Have the authors considered the possibility that the type II (page 5, line 133) and type III fibers in the walking muscles of the coxa are the same? Based on the images provided in Figure 1k and Supp. Figure 1c, the sarcomeres stained for F-actin show no discernible H zones, actin-free regions of the sarcomere, as seen in Figures 1a, 1d and 1i. The lack of a H-zone suggests that the muscles are contracted and hence

appear different from the Type III fibers, making the type II fibers an artifact. Experiments to address this should be conducted.

Response: We thank the reviewer for these comments. We have now quantified sarcomere distance in all fiber types and we find that there is significant difference among them as suggested by the reviewer. However, we now show that the mitochondrial networks in Fiber II but not Fiber I or III are altered in *cut* KD flies demonstrating that Fiber II is not the same as Fiber III.

C. Experimental design I: The authors use mito-gfp in some situations and mito-mcherry in others. Are they the same? Also, it is accepted practice in the *Drosophila* field that controls mimic the experimental situation, so Mef2GAL4 X UAS-gfp (for example) as the control for Mef2GAL4X UAS-mito-gfp. Also the authors need to control for GAL4 dilution effect, as it isn't acceptable to compare UAS-mito-gfp to UAS-mito-gfp, Uas-Gene of interest. The control needs to have UAS-mito-gfp plus another UAS. This is important as the authors may not be getting as much knockdown or overexpression in these conditions, as they expect. QRT PCR, Westerns, or IF with quantification should be done to assess the knockdown and overexpression manipulations.

Response: We thank the reviewer for these comments. We would have liked to only use mito-GFP, which is matrix targeted, for all experiments, but we were unable to combine it with Salm OE. Thus, we used mito-mCherry, which is outer membrane targeted, for the Salm OE experiments. We have now provided additional images from flight, jump, and leg muscles showing mito-GFP and mito-mCherry together demonstrating that mitochondrial network configuration is similar for both fluorophores and also providing a double UAS control for the gene misexpression experiments. Additionally, immunohistochemical staining and fluorescence intensity quantification was performed to confirm the knock down and overexpression of proteins of interest.

D. The authors show data suggesting that there are different mitochondrial networks between muscle types. Within the tubular muscles (jump and leg) there are parallel (jump, type 1 leg) and grid (type 2, 3). However in subsequent text and most figures, there is "Leg-coxa" and the reader is unclear whether it is fiber type 2 or 3. The authors say "walking muscles" in the text and in some figures, but they can be precise and tell us which, and with measurements tell us the amount of transformation from one type of

organization (be it mitochondrial networks or myofibril organization). Hence it is difficult to assess the data and the conclusions drawn.

Response: We thank the reviewer for this comment and have clarified the specific leg fiber region at all instances in the text. We have also quantified mitochondrial network orientation for all wild type, knockdown, and overexpression muscles.

E. Proteomics. The proteomics analysis outlined in Figure 4 is not clearly explained. Validation of the proteomics study should be included. Western analysis can be done to show that known targets of Salm are appropriately regulated as suggested. In addition the authors conclude ,in Line 239, that “These results are consistent with salm as a negative regulator of most of its targets to maintain the fibrillar nature of the flight muscles in *Drosophila*.” How does this fit with the authors own genetic data (expression of Salm in some of the tubular muscle – which normally doesn’t express Salm-- converts those muscles to fibrillar? Moreover it doesn’t fit with published data from the Schnorrer group. This needs clarification.

Response: We thank the reviewer for these comments. We have now added a better description of the proteomic analysis including the rationale behind it. Additionally, we performed quantitative immunofluorescence of H15 in Salm KD flight muscles which shows a decrease in abundance compared to wild type flight muscles in line with the proteomic results.

Spalt related proteins including salm itself have previously been shown to act as transcriptional repressors in *Drosophila* (Sanchez et al. *Drosophila Sal and Salr are transcriptional repressors. Biochemical Journal* 438.3 (2011): 437-445) and in mammalian cells (Netzer et al. *Human Molecular Genetics*, 2001), and we are unaware of data from Schnorrer or anyone showing data on direct interactions between salm and specific targets in muscle cells to indicate the directionality of salm activity. However, since this point is not critical to the outcome or interpretations here, we have removed it from the manuscript.

D. H15 knockdown. As these experiments are important for the author’s major conclusions, the amount of kd must be quantified in the muscle types. The authors conclude from their data that H15 is necessary for the grid like arrays in some leg muscles. The data shown , however, are not convincing. It looks like there are fewer mitochondria and as a result, the parallel arrays are not readily observable

and the authors interpret the pattern as a grid array. Again, TEM data would support the point that the authors wish to make. Not clear why overexpression studies were not included to make a point of sufficiency.

Response: We thank the reviewer for this comment. Immunohistochemical staining and fluorescence intensity quantification has now been performed to confirm the knock down and overexpression of proteins of interest. We have quantified mitochondrial volume as well as mitochondrial network orientation to confirm the phenotype in all conditions. We also performed *H15* over expression under *salm* KD condition and showed that the restoration of H15 expression reverses the effects of *salm* KD on muscle fiber type.

E. Figure prep: All figures can be significantly improved. When showing individual channels, greyscale images should be used for clarity. Better images should be possible. High magnification insets have been provided for some panels. These insets should be provided for all the panels along with scale bars. Similarly, deciphering scale bar size from the images is nearly impossible and should be mentioned in the Figure legends. This is true for all scale bars and high-magnification insets throughout the paper.

Response: The figures have been extensively modified to address these suggestions including using greyscale for individual channels and including scale bars and their size in the figure legends.

3. Writing – the authors should carefully re-read the manuscript as there are issues of clarity throughout. The introduction and discussion are long-winded, and not arranged logically. There are numerous grammatical mistakes throughout the paper. Moreover they should carefully consider their word choice, as the data shown do not show what they conclude.

Response: We thank the reviewer for these suggestions. We have now almost entirely re-written the manuscript including significantly shortening both the introduction and discussion. We believe the revised manuscript is more clear, free of grammatical errors, and the conclusions more directly supported by the now quantitative data.

Listed below are some of the concerns regarding the language and grammar used in the manuscript.

Some examples:

The abstract is drastically over the word limit at 248 words.

Response: The abstract is now within the specified word limit

Mitochondria is plural. Individual mitochondria are called mitochondrion. please italicize *Drosophila* whenever used in the paper. The nomenclature in *Drosophila* is: *Salm*: protein and *salm*<: gene.

Response: We have made the appropriate corrections for nomenclature as well as using mitochondrion when referring to an individual entity (e.g. a single individual mitochondrion) and mitochondria when referring to many entities (e.g. all individual mitochondria).

Line 43: Misexpression of ...spalt major (*salm*) converted the fibrillar flight muscles to tubular.... Loss of *salm* converts fibrillar to tubular. Gain of *salm* converts tubular to fibrillar. The sentence is not clear.

Response: This sentence has been removed in the rewritten manuscript.

Line 80. Myogenic factors, like myostatin, MyoD, Mef2, and calcineurin These are all different types of proteins, including signals(myostatin) and transcriptional regulators (MyoD and Mef2). Best to revise this sentence to clarify for a general audience.

Response: This sentence has been removed in the rewritten manuscript.

Line 98 neurogenic?

Response: This sentence has been removed in the rewritten manuscript.

Line 161- "flighted" is not a valid word to describe the flight capacity of flies.

Response: We have removed use of this word.

Line 293: different mitochondrial networks in *Drosophila* muscles are reflective of functional properties of the muscle fibers that they reside in. This is a bit of an overinterpretation as no functional aspects were shown here.

Response: This sentence has been removed in the rewritten manuscript. However, the functional differences among the flight, jump, and leg muscles have been described previously (e.g. Elliot and Sparrow. *Methods* 56(1), 2012 and Swank. *Methods* 56(1), 2012) and these citations are now included in the introduction and discussion.

Line 302. 'Walking muscles are the most frequently used muscle group " data? Reference?

Response: We have removed this sentence.

Line 305: "...walking muscles are comprised of heterogenous types of muscle fibers..." The authors haven't shown in this manuscript that the leg muscles (figure 1) are composed of different fiber types. Moreover, are the authors suggesting that each "group" (leg muscles 1, 2, 3) is made up of different myofiber types like one see in the TA muscle of mammals?

Response: In response to the reviewers comments, we have quantified sarcomere distance in all fiber types and we find that there is significant difference among them. Mitochondrial content and network orientation has also been quantified highlighting the differences between Fiber I and Fibers II and III. Additionally, mitochondrial networks in Fiber I but not Fibers II and III were affected by H15 KD, and mitochondrial networks in Fiber II but not Fibers I and III were affected by cut KD further suggesting that each of these three leg fiber regions are regulated differently.

Line 321. Since there is no quantification, the authors really can't draw this conclusion. Later in this para, "abnormal individual mitochondrial morphology". Again quantification would support this. If done.

Response: We thank the reviewer for this comment and have now quantified mitochondrial volume, mitochondrial number, mitochondrial aspect ratio and mitochondrial network orientation in all conditions to provide support to our conclusions.

Line 335, need to introduce, exd, hth, vg ..these symbols don't mean a lot to the general reader.

Response: We thank the reviewer for pointing this out. We have now introduced the full names of these factors when introduced in the last paragraph of the discussion.

REVIEWER COMMENTS

Reviewer #1 (Remarks to the Author):

The authors have done wonderful job to answer the queries. The paper can now be accepted.

Reviewer #2 (Remarks to the Author):

The authors have addressed all the points and concerns raised previously, which have improved the manuscript significantly.

Minor: the authors should use TEAB in line 579 and 582 in stead of triethyl ammonium bicarbonate.

Reviewer #4 (Remarks to the Author):

I have only been called in by the editor at the revision stage to comment on the fly specific aspects of former reviewer 3. I tried to do this, but also give some other comments that I believe are important to further improve this manuscript.

Katti, Glancy and colleagues are investigating here how different mitochondrial network morphologies present in different muscle types in the *Drosophila* model are regulated. This is certainly an interesting and unsolved question in the field of muscle and mitochondria biology.

Here the authors are investigating *Drosophila* flight muscle and leg muscle mitochondria morphologies, reproduce that *Salm* is the flight muscle selector gene regulating both myofibril and mitochondria morphology and following differential proteomics investigate a potential role of *H15* and *cut* in muscle fate choice.

Although the general approach is interesting, in my opinion many of the conclusions drawn by the authors are not supported by the currently provided data. In particular the main conclusion that mitochondria and myofibril morphology are regulated independently is only weakly supported.

Major comments:

1. I do not understand why the authors stress, already in abstract, to only search for pathways that regulate mitochondria networks independent of the 'contractile fiber type'. Naturally, they should be regulated together, and recent evidence showed that changing one, feedbacks to change the transcriptional fate of the other one (PMID 33828099).

Mammalian glycolytic or oxidative fibers also express different contractile proteins (e.g. different MyHCs, hence their names). Thus, they have not only different metabolic phenotypes (PMID 22013216). Please amend the statement in the introduction.

2. I do not think it is wise to call '*cut*, *salm*, and *H15*' an "evolutionarily conserved pathway". These genes code for three conserved transcription factors which are not a pathway. Hence, I also strongly recommend to change the title.

3. First results section: "Independent regulation of mitochondrial network and contractile types in *Drosophila* muscles". I do not see that this section does provide evidence for the strong statement made in the title of the section.

4. The strong claim that different leg muscle fiber types (I, II and III) exist in the coxa is in my opinion not supported by the data presented. It is well known by published data (PMID 33828099 and visible by the 3D reconstructions shown by the authors in Figure 1r and 1s that the mitochondrial network appears differently at different sections of these tubular fibers. However, from these reconstructions I cannot see the obvious differences the authors refer to in the single sections shown

in Figure 1l – q. To support the claim, I would like to see 3D reconstructed images of the three different fiber types. Different leg muscles do have different names (see PMID 15537687 for a modern reproduction of Miller’s 1950 nomenclature), which muscles would be type I, II, III? To my knowledge no differentially expressed genes are known that may cause differences in sarcomere length depending on fiber type. The here reported ‘differences’ in the sarcomere length might be easily explained by differences in the contraction-relaxation-cycle, which will vary depending on the leg positions between samples (supported by the very varying sarcomere length reported here for type III in Figure S3f). To substantiate differences in sarcomere length, absolute thin or thick filament length, which are characteristic to muscle types, would need to be measured. Thus, the conclusion that “contractile and metabolic phenotypes can be regulated independently in *Drosophila* muscles” is not supported by the data shown.

5. Salm mis-expression in leg muscles can cause a transformation of the tubular to partially fibrillar muscle fiber phenotype. However, this transformation is partial depending on the genetics used (see PMID 22094701). The same is seen by the authors here. There is only a partial change of the myofibrils, as least in the image shown in Figure 2l, which is very different to 2c. However, the authors claim that this transformation is complete on the myofibril level and hence conclude the only partial (or missing?) change in mitochondrial morphology is surprising. Again, these data are not convincing to argue that mitochondrial and myofibril morphology are regulated differentially.

6. Proteomics: I am surprised that the ‘classical’ flight muscle specific proteins (TpnC4, Act88F, Fln, Strn-MLCK, Aret) seem not to be included in the 142 proteins found to be fiber/mitochondria type specific. Any reason for this? It would help if the Flybase gene names are listed for all genes in the Table S2 and not only the protein ID code. All these genes need Salm for its expression (PMID 25532219), so why do the authors here find only H15 to positively depend on Salm? Were all the others not found in the mass spec analysis?

7. H15 function: Please report throughout the paper which H15 RNAi was used in the respective experiments (and not only in the methods V28415, V106875). Were flightlessness and fiber phenotypes found with both when driven with Mef2-GAL4? This needs careful documentation to rule out any off-target effects.

8. H15 effect on flight muscle myofibrils: How complete is the tubular switch? Again, without a more careful analysis of the expression of flight muscle specific sarcomeric proteins (TpnC4, Act88F, Fln, Strn-MLCK) a claim that the fiber change to tubular fate but the mitochondria do not is only weakly supported. The present fibers shown in Figure 4d are certainly not normal, but if they are ‘tubular’ I cannot judge. This would at least need a 3 D reconstruction and quantify the central location of the nuclei in the ‘transformed’ muscle, as seen in tubular fibers. (Figure S15h, seem to show that the nuclei are not in the fiber center in the H15 RNAi, hence there is no tubular transformation, weakening the claim of the authors).

9. H15 effect on leg muscle mitochondria: see my comment above on the type I, II, III types. Without careful 3D quantification of the different fiber types (or any other specific markers), not convincing to this reviewer.

10. H15 OE rescue of Salm RNAi: As mentioned before by another reviewer, this experiment needs a GAL4 dilution control, which is not obvious from the figure or the legend provided. Please include a second UAS in the Salm RNAi genotype with Mef2-GAL4. It would help if the genotypes used in all the experiments would be more clearly specified.

11. Role of cut in mitochondria morphology in leg muscles: Again, this needs better 3D reconstruction to make the conversion from grid to parallel in type II convincing.

12. cut, salm double RNAi: the GAL4 dilution control is missing again. In fact, Figure 6u shows that in

this 'double knock-down' Salm protein is actually not knocked-down. Furthermore, there does not seem to be more Salm protein in cut knock-down IFMs. Thus, why are the authors showing these non conclusive data, to generate a confusing hypothesis of cut being a negative regulator of salm? As the authors cite correctly vg and not cut (as they claim now) is the upstream Salm regulator (PMID 11740944). Is vg changed?

Minor comments:

1. abstract: "Here, we show that natural energetic demands placed on *Drosophila melanogaster* muscles yield native cell-types among which contractile and mitochondrial network-types are regulated independently." I do not understand this sentence. If it means that mitochondria and myofibril morphologies are regulated differentially, I would like to see more data supporting this strong claim in the paper.

2. Reported ZASP Cherry allele. This is supposedly a Cherry integration into a ZASP52 MiMIC line, so not a UAS line, as currently reported in the methods. I suppose the authors mean Zasp52MI02988 as MI02908 is inserted in a gene called Fili.

3. A few typos: *Drosophila* should be italics and capital also in the methods. Frieder Schoeck is misspelled. Phalloidin or phalloidin? jump muscle should not be capital letter.

RESPONSE TO REVIEWER COMMENTS

Reviewer #1 (Remarks to the Author):

The authors have done wonderful job to answer the queries. The paper can now be accepted.

We thank Reviewer #1 for their original queries which led to significant improvement in this work.

Reviewer #2 (Remarks to the Author):

The authors have addressed all the points and concerns raised previously, which have improved the manuscript significantly.

We thank the reviewer for their previous questions/comments which resulted in a greatly improved manuscript.

Minor: the authors should use TEAB in line 579 and 582 in stead of triethyl ammonium bicarbonate.

We have made the suggested change.

Reviewer #4 (Remarks to the Author):

I have only been called in by the editor at the revision stage to comment on the fly specific aspects of former reviewer 3. I tried to do this, but also give some other comments that I believe are important to further improve this manuscript.

We thank the reviewer for stepping in at this stage and for their comments which have resulted in an improved manuscript.

Katti, Glancy and colleagues are investigating here how different mitochondrial network morphologies present in different muscle types in the *Drosophila* model are regulated. This is certainly an interesting and unsolved question in the field of muscle and mitochondria biology.

Here the authors are investigating *Drosophila* flight muscle and leg muscle mitochondria morphologies, reproduce that *Salm* is the flight muscle selector gene regulating both myofibril and mitochondria morphology and following differential proteomics investigate a potential role of H15 and cut in muscle fate choice.

Although the general approach is interesting, in my opinion many of the conclusions drawn by the authors are not supported by the currently provided data. In particular the main conclusion that mitochondria and myofibril morphology are regulated independently is only weakly supported.

We thank the reviewer for their interest in our work and for pointing out several areas where we have now improved clarity of our data and interpretations. Regarding the independence of mitochondria and myofibrillar morphology, we have provided nine different examples of mitochondrial morphology being altered without changing the fibrillar or tubular nature of the muscle (jump vs leg, leg Fiber 1 vs

leg Fiber II/III, Marf KD flight vs WT, Fis1 KD flight vs WT, Drp1 KD flight vs WT, Miro KD flight vs WT, H15 KD jump vs WT, H15 KD leg Fiber I vs WT, cut KD leg Fiber II vs WT) and now include several videos of 3D renderings to better support these conclusions as suggested by the reviewer.

Additionally, as requested by the reviewer, we have now rewritten the Introduction to better clarify the rationale for our work. We now discuss and cite the many papers over several decades which have shown that metabolic fiber type and contractile fiber type are intertwined but separate processes in mammalian muscles. We hypothesized that this well-established premise in mammalian muscle biology would also be true among *Drosophila* tubular muscles thereby allowing us to take advantage of fly genetics to search for regulators of the fiber type specification process. Indeed, we find that tubular muscles can take at least two different mitochondrial network configurations with the most striking difference being between the parallel networks in the jump muscle and the grid-like networks in the leg muscle.

Major comments:

1. I do not understand why the authors stress, already in abstract, to only search for pathways that regulate mitochondria networks independent of the 'contractile fiber type'. Naturally, they should be regulated together, and recent evidence showed that changing one, feeds back to change the transcriptional fate of the other one (PMID 33828099).

Mammalian glycolytic or oxidative fibers also express different contractile proteins (e.g. different MyHCs, hence their names). Thus, they have not only different metabolic phenotypes (PMID 22013216). Please amend the statement in the introduction.

We thank the reviewer for this comment and agree that there are plenty of studies showing that both the contractile type and mitochondrial metabolism can be regulated together. However, there is also a lot of data available demonstrating that mitochondrial metabolism/morphology can be altered without changing contractile type. For example, loss of the master transcriptional regulator of mitochondrial biogenesis, PGC1 α , in mouse striated muscle led to deficits in mitochondrial metabolism and a reduction in mitochondrial dynamics proteins without altering the contractile fiber type (PMID: 16054070 & 21109195). Similarly, in flies, loss of PGC1 α ortholog, spargel, leads to altered oxidative metabolism and mitochondrial morphology but no change in the fibrillar nature of the flight muscles (PMID: 28407521). This could not occur if mitochondrial morphology was always regulated together with contractile type.

Even PMID 33828099 as mentioned by the reviewer provides evidence for separate regulation of mitochondrial morphology and contractile type in *Drosophila* muscles as Marf knockdown flight muscles have a reduction in mitochondrial size without altering the fibrillar nature of flight muscles. There are many other studies, including this one, also showing that individual mitochondrial size can be increased or decreased without altering the fibrillar nature of *Drosophila* flight muscles (PMIDs: 33450208, 24198395, 18799731, 29742430, 30354903, 34061429, 25376463, 27716788). This could not occur if mitochondrial morphology was only regulated together with contractile type. Additionally, recent work in mice (PMID: 35150906) shows that skeletal muscle specific loss of Cpt2 causes a change

in metabolic fiber type without a change in myosin heavy chain isoforms. Thus, there is clear data from many labs showing that mitochondrial content, size, and metabolism can be regulated without altering contractile type across species, and several specific regulators of these processes are known. However, there is very little information regarding the regulation of another major component of mitochondrial network architecture, where mitochondria are placed in a muscle cell. While we are very interested in the integrated aspects of how mitochondrial location impacts the contractile apparatus (see Katti et al. bioRxiv, 2022), we hypothesized here that regulation of mitochondrial location can also diverge from contractile type regulation similar to the above mentioned studies regarding mitochondrial content and size. We have now amended the introduction to more clearly state our rationale for this work and have included citations to many of the available papers showing that mitochondrial morphology can be altered without affecting contractile fiber type.

2. I do not think it is wise to call 'cut, salm, and H15' an "evolutionarily conserved pathway". These genes code for three conserved transcription factors which are not a pathway. Hence, I also strongly recommend to change the title.

We thank the reviewer for this suggestion and have now removed "pathway" from the title.

3. First results section: "Independent regulation of mitochondrial network and contractile types in *Drosophila* muscles". I do not see that this section does provide evidence for the strong statement made in the title of the section.

We thank the reviewer for this comment. As now discussed in the Introduction, *Drosophila melanogaster* muscles are classified into either fibrillar or tubular contractile types. In Figure 1, we show that the jump muscle has parallel mitochondrial networks and that leg muscles can have grid-like mitochondrial networks despite each muscle having a tubular contractile type with the same primary actin isoform (Actin79B from PMIDs: 29318731, 6432334). These data clearly show that different mitochondrial network configurations can occur in tubular muscles. Thus, mitochondrial network configuration can be regulated independently within tubular muscles. Similarly, Fiber I in the leg muscles has parallel mitochondrial networks (now supported with 3D reconstructions) whereas Fibers 2 and 3 have grid-like mitochondrial networks despite all muscles being tubular. Again, these data show different mitochondrial network configurations are possible within the same contractile type. Additionally, the flight and jump muscles both have parallel mitochondrial networks despite the differences in contractile type. Thus, similar mitochondrial network configurations can occur in different contractile types. We have now added reference to the same primary actin isoforms among the jump and leg muscles to further strengthen this section.

4. The strong claim that different leg muscle fiber types (I, II and III) exist in the coxa is in my opinion not supported by the data presented. It is well known by published data (PMID 33828099 and visible by the 3D reconstructions shown by the authors in Figure 1r and 1s that the mitochondrial network appears differently at different sections of these tubular fibers. However, from these reconstructions I cannot see the obvious differences the authors refer to in the single sections shown in Figure 1l – q. To support the claim, I would like to see 3D reconstructed images of the three different fiber types.

Different leg muscles do have different names (see PMID 15537687 for a modern reproduction of Miller's 1950 nomenclature), which muscles would be type I, II, III?

To my knowledge no differentially expressed genes are known that may cause differences in sarcomere length depending on fiber type. The here reported 'differences' in the sarcomere length might be easily explained by differences in the contraction-relaxation-cycle, which will vary depending on the leg positions between samples (supported by the very varying sarcomere length reported here for type III in Figure S3f). To substantiate differences in sarcomere length, absolute thin or thick filament length, which are characteristic to muscle types, would need to be measured. Thus, the conclusion that "contractile and metabolic phenotypes can be regulated independently in *Drosophila* muscles" is not supported by the data shown.

We thank the reviewer for these suggestions and have now provided supplementary videos of the 3D renderings of the three different leg muscle fibers which provide stronger support for the differences in mitochondrial networks between fiber I and II,III. The pairs of mitochondria running perpendicularly to the muscle contractile apparatus adjacent to the Z-disk are not present in Fiber I but are found in Fibers II and III. Moreover, we find that H15 KD only acts on leg Fiber I, cut KD only acts on leg Fiber II, and salm OE only impacts Fiber I providing additional evidence that the three leg regions are regulated differently. Additionally, differences in leg muscle structures shown here are consistent with previous histochemical analyses finding two different metabolic phenotypes in the leg (PMID: 142843) as cited in the discussion. Thus, these data now more clearly show that both parallel and grid-like mitochondrial networks can occur in the tubular leg muscles.

Sarcomere lengths in each muscle are reported at the request of the previous reviewer #3 but we do not interpret these data to signify any differences in regulation between the different regions due to the possibilities mentioned by the reviewer.

It is difficult to match our leg locations to the data in the suggested paper due to the differences in the types of stainings used. However, it appears that our Fiber III is likely the trochanter reductor muscle. We are unable to precisely determine whether the trochanter rotator muscle or the trochanter levator muscles are Fiber I or Fiber II.

Both cut KD (Fiber II) and H15 KD (Fiber I) alter mitochondrial network configuration without altering the tubular nature of the contractile apparatus in the legs. Similarly, wild type Fiber I has parallel mitochondrial networks whereas wild type Fibers II and III have grid-like mitochondrial networks. These data are all now better supported by videos of the 3D renderings of the mitochondria in the leg muscles. These data clearly show differences in mitochondrial network configuration among muscles despite all having a tubular contractile type.

5. Salm mis-expression in leg muscles can cause a transformation of the tubular to partially fibrillar muscle fiber phenotype. However, this transformation is partial depending on the genetics used (see PMID 22094701). The same is seen by the authors here. There is only a partial change of the myofibrils, as least in the image shown in Figure 2I, which is very different to 2C. However, the authors claim that this transformation is complete on the myofibril level and hence conclude the only partial (or missing?) change in mitochondrial morphology is surprising. Again, these data are not convincing to argue that

mitochondrial and myofibril morphology are regulated differentially.

Unfortunately, we are unsure which statements the reviewer is referring to regarding a claim for complete transformation of the salm OE leg myofibrils. We stated that “salm OE transformed the tubular leg myofibrils to more fibrillar-like myofibrils” which is similar to, if not softer than, the terminology used in PMID 22094701 (“salm expression induces a clear transformation of the tubular leg muscles into fibrillar IFM-like muscles”). Indeed, the partial conversion of the Salm OE leg muscles we observe here is similar to the partial transformation in PMID 22094701 (our WT and Salm OE figures are in the upper panels below. Lower panels are from PMID 22094701). Thus, we never state that salm OE results in a complete transformation nor do we claim any surprising results or make any specific statements regarding independent regulation of mitochondrial and myofibril morphology in the Salm OE section.

6. Proteomics: I am surprised that the ‘classical’ flight muscle specific proteins (TpnC4, Act88F, Fln, Strn-MLCK, Aret) seem not to be included in the 142 proteins found to be fiber/mitochondria type specific. Any reason for this? It would help if the Flybase gene names are listed for all genes in the Table S2 and not only the protein ID code. All these genes need Salm for its expression (PMID 25532219), so why do the authors here find only H15 to positively depend on Salm? Were all the others not found in the mass spec analysis?

We thank the reviewer for this comment. TpnC4 was identified as positively associated with fibrillar muscles and is included in Supplemental Dataset 2 as one of the 142 proteins. Aret was not detected in our proteomic screen. Act88F was the most abundant actin isoform in flight muscles whereas

Act79B was the most abundant isoform in jump, leg, and Salm KD flight muscles as would be expected. However, the relative differences in expression of each isoform among muscles was not sufficient to meet the threshold of our screen. Fln and Strn-MLCK were highly upregulated in the WT flight muscles compared to TDT, leg, and salm KD flight muscles as would be expected. However, the expression of both Fln (1.43-fold increase) and Strn-MLCK (1.40-fold) was not upregulated sufficiently in the Salm OE leg muscle to meet the two-fold threshold used in our screen. We have now updated Table S2 with the Flybase gene names as suggested.

7. H15 function: Please report throughout the paper which H15 RNAi was used in the respective experiments (and not only in the methods V28415, V106875). Were flightlessness and fiber phenotypes found with both when driven with Mef2-GAL4? This needs careful documentation to rule out any off-target effects.

We thank the reviewer for this question and agree that this needs careful documentation. Initial tests were performed with both lines and found that V106875 resulted in weak flight ability whereas V28415 led to viable, flightless flies. Thus, we chose to pursue the stronger phenotype for downstream analyses. This has now been stated at the introduction of the H15 KD data in the Results.

To test whether the H15 KD phenotypes in the V28415 line were a result of off-target effects, we assessed the flight muscle contractile apparatus in RNAi KD lines for eight of the 16 off-target genes whereas the other eight genes had been tested previously for contractile phenotypes in a large scale genetic screen. None of these KD lines led to a tubular contractile apparatus in the flight muscle. These data are now included as Supplemental Table 1, also included below for the reviewer. These data suggest that off-target effects are not the cause of the H15 KD phenotypes observed. Further support for the specific role of H15 is provided by the H15 overexpression driven rescue of the tubular phenotype caused by salm KD. Thus, H15 KD causes a tubular switch in the flight muscles and H15 OE can return tubular flight muscles to fibrillar.

H15 OFF target (gene name)	Phenotype with Mef2 Gal4	Stock ID	Reference
CG8127	Weak Flyer	BS# 35780, BS# 43231	This Paper
CG16711	wild type		PMID: 20220848
CG32717	wild type	BS# 33909, BS# 33991	This Paper
CG7847	Lethal and weak flyer	BS# 27701	PMID: 20220848 and this Paper
CG7031	wild type		PMID: 20220848
CG43122	wild type	BS# 25995	PMID: 20220848, and This Paper
CG43749	wild type		PMID: 20220848
CG11202	Wildtype	BS# 62953	This Paper
CG13194	Weak Flyer	BS# 63547	This Paper
CG33988	wild type		PMID: 20220848
CG11711	Wild type		PMID: 20220848

CG31374	Lethal and Flight less	BS# 31548,	This Paper
CG10192	wild type		PMID: 20220848
CG34380	wild type		PMID: 20220848
CG6175	weak flyer	BS# 62516	PMID: 20220848 and This Paper
CG12443	Weak flyer		PMID: 20220848

8. H15 effect on flight muscle myofibrils: How complete is the tubular switch? Again, without a more careful analysis of the expression of flight muscle specific sarcomeric proteins (TpnC4, Act88F, Fln, Strn-MLCK) a claim that the fiber change to tubular fate but the mitochondria do not is only weakly supported. The present fibers shown in Figure 4d are certainly not normal, but if they are ‘tubular’ I cannot judge. This would at least need a 3 D reconstruction and quantify the central location of the nuclei in the ‘transformed’ muscle, as seen in tubular fibers. (Figure S15h, seem to show that the nuclei are not in the fiber center in the H15 RNAi, hence there is no tubular transformation, weakening the claim of the authors).

We thank the reviewer for the comment. *H15* KD results in the transformation of the fibrillar contractile type to the tubular type in Flight muscles (DLMs). We have now included in the manuscript the cross-section view and 3D rendering of the *H15* KD muscle fiber showing the tubular-like muscle fiber with the centralized nuclei (new Fig. S15). Further, in response to the reviewer’s suggestion, we have performed the qPCR quantitative analysis of fibrillar muscle-specific sarcomeric genes, *TpnC4*, *Act88F*, and *Fln*. In *H15* KD flight muscles, these sarcomeric genes were significantly decreased compared to the levels in wildtype flight muscles (included in Fig. S15). This data further strengthens our conclusion that *H15* regulates contractile type in flight muscles, particularly when combined with the *H15* OE rescue experiments.

Additionally, for the reviewers info, we find that peripherally located nuclei can sometimes occur in wild type tubular muscles. Below is a focused ion beam scanning electron microscopy image of a wild type tubular leg muscle with a peripherally located nucleus. The raw data for this dataset is available at <https://zenodo.org/record/5796264>.

9. H15 effect on leg muscle mitochondria: see my comment above on the type I, II, III types. Without careful 3D quantification of the different fiber types (or any other specific markers), not convincing to this reviewer.

We thank the reviewer for this comment. We have now included 3D rendered video of H15 KD fiber I which shows mitochondria now forming a grid-like network rather than a parallel network as in the wild type.

10. H15 OE rescue of Salm RNAi: As mentioned before by another reviewer, this experiment needs a GAL4 dilution control, which is not obvious from the figure or the legend provided. Please include a second UAS in the Salm RNAi genotype with Mef2-GAL4. It would help if the genotypes used in all the experiments would be more clearly specified.

We thank the reviewer for this suggestion. A second UAS in the Salm RNAi genotype with Mef2-Gal4 is shown in Figure 2f-h (Mito-GFP) and Figure S8 (KDEL-RFP). Importantly, Salm RNAi flight muscles with one UAS (Figure 5f) or two UAS (Figure 2f) show the same grid-like mitochondrial networks indicating that Gal4 dilution is not the cause of the Salm KD rescue by H15 OE. This control is now mentioned in the text of the results section describing the H15 OE rescue.

11. Role of cut in mitochondria morphology in leg muscles: Again, this needs better 3D reconstruction to make the conversion from grid to parallel in type II convincing.

We thank the reviewer for the comment. We have now included 3D video of *cut* KD fiber II mitochondrial networks (Supplementary video 8) as well as the wild type muscles as mentioned above.

12. *cut*, *salm* double RNAi: the GAL4 dilution control is missing again. In fact, Figure 6u shows that in this 'double knock-down' *Salm* protein is actually not knocked-down. Furthermore, there does not seem to be more *Salm* protein in *cut* knock-down IFMs. Thus, why are the authors showing these non conclusive data, to generate a confusing hypothesis of *cut* being a negative regulator of *salm*? As the authors cite correctly *vg* and not *cut* (as they claim now) is the upstream *Salm* regulator (PMID 11740944). Is *vg* changed?

We thank the reviewer for these comments. As discussed above, the Gal4 dilution control with a second UAS on the *Salm* RNAi background is shown in Figure 2f. This is now mentioned in the text when discussing the *cut*, *salm* double RNAi data. Indeed, we were surprised when *salm* protein was no longer knocked down in the *cut*: *salm* double KD flight muscles. Based on the mito-GFP UAS control suggesting this was not a Gal4 dilution effect, we hypothesized that *cut* KD was acting on *salm* expression rather than bypassing it the way the H15 OE rescue does. Indeed, we show that *cut* KD results in an increase in *salm* transcript expression in both the wild type and *salm* KD muscles. Additionally, these data suggesting that *cut* regulates *salm* expression are consistent with the known role of *cut* in the development of the tubular direct flight muscles where high *cut* expressing cells in the third instar larval stage go on to become direct flight muscles lacking *salm* and the low *cut* cells are able to proceed to the *salm* expressing fibrillar indirect flight muscles (PMID: 11740944).

As we cite and the reviewer mentions, *vg* is an upstream regulator of *salm*, but it is not the only one as *Hth* and *Exd* are also upstream regulators of *salm* (PMID: 22975331). Based on our data showing the *cut* KD rescue of the *salm* KD phenotype as well as the data showing the *cut* KD increases *salm* expression in a wild type background, we concluded that *cut* acts as a repressor of *salm*. However, while the data discussed above find that *vg*, *hth*, *exd*, and *cut* all can regulate *salm* expression, we are unaware of any data showing that any of these genes act directly on *salm*, and we do not rule out that *cut* repression of *salm* is mediated through *vg*. In fact, *cut* and *vg* mutually repress each other (PMID: 11740944), so it is also possible that *vg* activates *salm* through repressing *cut*.

To clarify the impact of *cut* expression on *salm*, we now demonstrate that *cut* overexpression driven by both *Act88F-Gal4* and *1151-Gal4* results in a tubular conversion of the IFMs together with changing the mitochondrial networks to a grid-like configuration. This occurs together with a loss of *salm* at the transcript and protein levels as well as a loss of H15 transcripts. Thus, *cut* KD increases *salm* expression and *cut* OE reduces *salm* expression consistent with *cut* acting as a repressor of *salm*.

Minor comments:

1. abstract: "Here, we show that natural energetic demands placed on *Drosophila melanogaster* muscles yield native cell-types among which contractile and mitochondrial network-types are regulated independently." I do not understand this sentence. If it means that

mitochondria and myofibril morphologies are regulated differentially, I would like to see more data supporting this strong claim in the paper.

We have now amended this sentence to state that mitochondrial and myofibril morphologies are regulated differentially. The additional data and text supporting this statement are discussed thoroughly in the responses above.

2. Reported ZASP Cherry allele. This is supposedly a Cherry integration into a ZASP52 MiMIC line, so not a UAS line, as currently reported in the methods. I suppose the authors mean Zasp52MI02988 as MI02908 is inserted in a gene called Fili.

We thank the reviewer for catching this error and have corrected the methods text.

3. A few typos: *Drosophila* should be italics and capital also in the methods. Frieder Schoeck is misspelled. Phalloidin or phalloidin? jump muscle should not be capital letter.

We thank the reviewer for catching these typos and have now corrected them.

REVIEWERS' COMMENTS

Reviewer #4 (Remarks to the Author):

I have to say that I am somewhat irritated by the strategy of the authors publishing two very related papers, even in the same journal, in parallel, without openly informing the reviewers about this. Even more disturbing is that the second paper, which is now published Ajayi et al. Nat Comm 2022 is addressing the sarcomere and myofibril structure (only) in flight, jump and leg muscles using state of the art 3D-reconstruction, without mentioning the mitochondria at all, whereas the currently manuscript did not provide these high quality reconstructions but continues to claim that both are regulated independently throughout.

In a way paper 2 scooped paper 1, and also contradicts it.

In the current paper under review Katti et al. conclude in the abstract that "H15 regulates mitochondrial network configuration but not contractile type in jump and leg muscles", a point that I discussed at length in my initial review as not well supported. Now, I read in the new Ajayi et al. Nat Comm 2022 "we demonstrate that the muscle-specific loss of cell-type specification factor H15 leads to the conversion of the jump muscles to a leg muscle-like phenotype with increased sarcomere branching and smaller myofibrils".

To me this is a clear contradiction to the current interpretation of the authors. Of course, I agree with the obvious interpretation that mitochondrial networks are not the same in all tubular muscle types. However, as the SAME authors beautifully document in Ajayi et al. Nat Comm 2022 the myofibrils are not the same either! More branching or less branching amongst other differences.

A convincing and rigorous analysis of this would provide the same quality level of mitochondria reconstructions together with myofibrils as done for myofibrils in Ajayi et al. Nat Comm 2022, for the type 1, type2 and type3 leg muscles, of wild type and the various RNAi lines used here. However, this happened only at the third round of review and even not to the same quality as done in Ajayi et al. Nat Comm 2022.

With my comments I was trying to help the authors to make their manuscript more rigorous, to avoid over-interpretations and to highlight claims that are not valid for me when seeing the supporting data. I am happy to see a new title and an entirely work-over intro. Of course I have no doubt that mitochondria and contractile structures can be regulated independently. Still, I find many of the interpretations in this revised paper strong over-statements, which in some cases, see above, contradict (in my opinion) the authors own statements in the parallel paper. I list some below.

However, I feel I have invested enough time and leave it to the editor to make a decision on this case.

1. end of intro " Our findings suggest that evolutionarily conserved transcription factors including cut, salm, and H15 regulate mitochondrial network configuration in muscle cells through a specification process which can operate independently of contractile type, mitochondrial content, and mitochondrial size." I strongly disagree with this statement, as cut and salm change fiber type fate, as shown by many including the authors here again, and H15 see Ajayi et al. Nat Comm 2022. Such bold conclusions are not helpful to support arguments that both mitochondria and contractile structures can be regulated independently, which I do not doubt.

2. Why are the beautiful new mitochondria reconstructions not used for the figures but readers are force to open all the supplements to see these cool data? why are the myofibrils disconnected in these reconstructions in flight, jump and leg muscles? this looks entirely different compared Ajayi et al. Nat Comm 2022 that concluded branched networks of fibrils.

3. Why is the difference in sarcomere length still included as an argument in leg fiber types when the

authors themselves are not convinced by it, see response letter.

4. The 3D reconstructions show that the myofibrils are not the same in type I, II and III legs, despite the caveats with the reconstructions, still the authors conclude that “demonstrating that contractile and metabolic phenotypes can be regulated independently in *Drosophila* muscles”. Just to repeat myself, mitochondria are different in type I, II and III, and myofibril morphology is, too.

5. Conclusion of the H15 flight muscle part “Thus, in contrast to salm KD, H15 KD induced a jump muscle phenotype (tubular/parallel) rather than a leg muscle phenotype (tubular/grid) in the flight muscles indicating that mitochondrial network configuration can be regulated independently from contractile type in the flight muscle.” The authors showed beautifully that jump and leg muscle myofibrils are not that same, but both are tubular, however they did not quantify the type of tubular transformation after H15 RNAi in IFM, so how can one conclude this?

Response to Reviewers Comments

Reviewer #4 (Remarks to the Author):

I have to say that I am somewhat irritated by the strategy of the authors publishing two very related papers, even in the same journal, in parallel, without openly informing the reviewers about this. Even more disturbing is that the second paper, which is now published Ajayi et al. Nat Comm 2022 is addressing the sarcomere and myofibril structure (only) in flight, jump and leg muscles using state of the art 3D-reconstruction, without mentioning the mitochondria at all, whereas the currently manuscript did not provide these high quality reconstructions but continues to claim that both are regulated independently throughout.

In a way paper 2 scooped paper 1, and also contradicts it.

In the current paper under review Katti et al. conclude in the abstract that “H15 regulates mitochondrial network configuration but not contractile type in jump and leg muscles”, a point that I discussed at length in my initial review as not well supported. Now, I read in the new Ajayi et al. Nat Comm 2022 “we demonstrate that the muscle-specific loss of cell-type specification factor H15 leads to the conversion of the jump muscles to a leg muscle-like phenotype with increased sarcomere branching and smaller myofibrils”.

To me this is a clear contradiction to the current interpretation of the authors. Of course, I agree with the obvious interpretation that mitochondrial networks are not the same in all tubular muscle types. However, as the SAME authors beautifully document in Ajayi et al. Nat Comm 2022 the myofibrils are not the same either! More branching or less branching amongst other differences.

We appreciate the reviewer’s concerns and assure the reviewer that parallel papers was not our intended strategy. This paper was first submitted to Nature Communications in February of 2020 and sent out for review before even a single dataset in Ajayi et al. was collected. In fact, this paper, combined with Willingham et al. (Nature Communications, 2020) serves as the basis for the deeper dive in Ajayi et al. However, this paper has been in review/revision for over 2.5 years whereas Ajayi et al. was published ~7 months after initial submission. Much of the delays with this paper have been due to the pandemic which restricted us to 50% or less lab occupancy for 1.5 years. The similarities and differences among these two papers and also Katti et al. bioRxiv, 2022 (just accepted at Nature Communications) were discussed previously with the editor, Dr. Bardot, who has been handling each manuscript.

This paper uses the classifications commonly used throughout the Drosophila muscle literature, fibrillar and tubular, to define contractile type. This paper shows, largely using light microscopy, that mitochondrial networks can take different configurations within tubular muscles and that H15 and cut can regulate this process. In other words, mitochondrial networks can change without altering the contractile type as commonly defined. Certainly, there are sub-types within the tubular contractile type (as shown in Ajayi et al. and elsewhere), but there are not well defined terms for the contractile sub-types other than using the anatomical location or muscle name. As such, we have stuck to the classical definition of contractile type here, fibrillar or tubular, as outlined in the introduction. We have now stated our use of these definitions more clearly in the introduction as well as the fact that contractile properties can differ among tubular muscles despite them being the same contractile type. Additionally, we have added text in the results describing how H15 KD in jump muscles does not

change the tubular contractile type but does change the myofibril network structure and myofilament structure in our other papers. All of these data are consistent with our original conclusion here that H15 KD in the jump muscle results in a more leg-like phenotype, though both the jump and leg muscles have the same contractile type (i.e. tubular).

A convincing and rigorous analysis of this would provide the same quality level of mitochondria reconstructions together with myofibrils as done for myofibrils in Ajayi et al. Nat Comm 2022, for the type 1, type2 and type3 leg muscles, of wild type and the various RNAi lines used here. However, this happened only at the third round of review and even not to the same quality as done in Ajayi et al. Nat Comm 2022.

We agree that EM reconstructions provide better resolution than the light microscopic images and renderings shown here. However, performing 3D EM on the nearly 90 different muscle types assessed in this paper (5 muscles: flight, jump, and three leg regions and 18 genotypes: WT, salm KD, salm OE, H15 KD, H15 OE, salm KD/H15 OE, cut KD, cut OE, salm KD/cut KD, pros KD, pros OE, lmpt KD, lmpt OE, drp1 KD, marf KD, fis1 KD, miro KD, and mid KD fly lines) is not feasible due to the relatively slow nature of electron microscopy where it takes 10-14 days from tissue dissection and fixation to image collection with image analysis taking additional days or weeks depending the types of analyses chosen. At our current rate of 3D EM dataset collection (among the highest in the world for muscle cells over the past five years), we collect ~20 datasets per year. Thus, it would take more than four years to collect the all data as proposed by the reviewer, and that would be for only a single biological replicate of each muscle type. Light microscopy is nearly an order of magnitude faster than EM and provides sufficient resolution to determine the primary outcomes measured here: whether the mitochondria are located in the parallel or perpendicular axis and whether the contractile apparatus is fibrillar or tubular.

With my comments I was trying to help the authors to make their manuscript more rigorous, to avoid over-interpretations and to highlight claims that are not valid for me when seeing the supporting data. I am happy to see a new title and an entirely work-over intro. Of course I have no doubt that mitochondria and contractile structures can be regulated independently. Still, I find many of the interpretations in this revised paper strong over-statements, which in some cases, see above, contradict (in my opinion) the authors own statements in the parallel paper. I list some below.

However, I feel I have invested enough time and leave it to the editor to make a decision on this case. **We thank the reviewer for stepping in after this work had already been in review/revision for more than 1.5 years. We think that the additional clarification that we are defining contractile type based on the classical literature definition while also mentioning that there may be differences in contractile properties among muscles of the same type (i.e. tubular) should ease the reviewer's concerns about over-statements potentially confusing readers.**

1. end of intro " Our findings suggest that evolutionarily conserved transcription factors including cut, salm, and H15 regulate mitochondrial network configuration in muscle cells through a specification process which can operate independently of contractile type, mitochondrial content, and mitochondrial size." I strongly disagree with this statement, as cut and salm change fiber type fate, as shown by many

including the authors here again, and H15 see Ajayi et al. Nat Comm 2022. Such bold conclusions are not helpful to support arguments that both mitochondria and contractile structures can be regulated independently, which I do not doubt.

As also stated above and in the introduction as revised previously, contractile type in flies has long been defined as either fibrillar or tubular, and this is the definition we are using in this paper. We have now stated this directly in the introduction while also making sure to mention that while there may be differences in specific contractile properties among the various tubular muscles, they are all the same contractile type based on the well-established fibrillar/tubular classification system.

2. Why are the beautiful new mitochondria reconstructions not used for the figures but readers are forced to open all the supplements to see these cool data? why are the myofibrils disconnected in these reconstructions in flight, jump and leg muscles? this looks entirely different compared Ajayi et al. Nat Comm 2022 that concluded branched networks of fibrils.

We thank the reviewer for these questions. The 3D renderings are included as Supplemental videos so that the reader can see how they look from many different angles in rotation rather than a single angle if they were used as single images. These 3D renderings look different than in Ajayi et al. because they come from confocal microscopy image stacks of phalloidin (i.e. actin) stained muscles. Since actin is not present throughout the entire length of the sarcomere, there are gaps where actin is not present. Conversely, Ajayi et al. performed 3D electron microscopy which has ~10 fold greater resolution and where the entire length of the sarcomere is visible and can be segmented to make the renderings shown in that work. In other words, different imaging technologies and different staining techniques explain the differences.

3. Why is the difference in sarcomere length still included as an argument in leg fiber types when the authors themselves are not convinced by it, see response letter.

While this data was included at the request of the previous reviewer #3, we have now removed this data from the paper.

4. The 3D reconstructions show that the myofibrils are not the same in type I, II and III legs, despite the caveats with the reconstructions, still the authors conclude that “demonstrating that contractile and metabolic phenotypes can be regulated independently in *Drosophila* muscles”. Just to repeat myself, mitochondria are different in type I, II and III, and myofibril morphology is, too.

While it is possible and perhaps likely that there are contractile differences among the three leg regions, all three regions are clearly tubular muscles. Thus, they are all the same contractile type as has been classically defined. As mentioned in response to point #1 above, we have now provided further clarification in the introduction that contractile type in this work means the classical definition.

5. Conclusion of the H15 flight muscle part “Thus, in contrast to salm KD, H15 KD induced a jump muscle phenotype (tubular/parallel) rather than a leg muscle phenotype (tubular/grid) in the flight muscles indicating that mitochondrial network configuration can be regulated independently from contractile type in the flight muscle.” The authors showed beautifully that jump and leg muscle myofibrils are not

that same, but both are tubular, however they did not quantify the type of tubular transformation after H15 RNAi in IFM, so how can one conclude this?

As mentioned above, contractile type here refers to either the fibrillar or tubular nature of the muscle. In the case of H15 KD in IFM, the contractile apparatus changes from fibrillar to tubular while the mitochondria remain in parallel. Thus, the contractile type changed while mitochondrial location did not indicating that these two processes can be regulated separately.